# The Role of Amphibian AMPs Against Oxidative Stress and Related Diseases

**DOI:** 10.3390/antibiotics14020126

**Published:** 2025-01-25

**Authors:** Yudy Lorena Silva Ortíz, Thaís Campos de Sousa, Natália Elisabeth Kruklis, Paula Galeano García, José Brango-Vanegas, Marcelo Henrique Soller Ramada, Octávio Luiz Franco

**Affiliations:** 1Grupo de Investigación en Productos Naturales Amazónicos (GIPRONAZ), Facultad de Ciencias Básicas, Universidad de la Amazonia, Florencia 180001, Caquetá, Colombia; yu.silva@udla.edu.co (Y.L.S.O.); p.galeano@udla.edu.co (P.G.G.); 2Graduate Program in Genomic Sciences and Biotechnology, Catholic University of Brasília, Brasília 71966-160, DF, Brazil; s.thaiscampos@gmail.com (T.C.d.S.); kruklisnatalia.24@gmail.com (N.E.K.); marceloramada@gmail.com (M.H.S.R.); 3Center for Proteomic and Biochemical Analyses, Graduate Program in Genomic Sciences and Biotechnology, Catholic University of Brasília, Brasília 70790-160, DF, Brazil; josbrang@gmail.com; 4S-Inova Biotech, Graduate Program in Biotechnology, Dom Bosco Catholic University, Campo Grande 79117-900, MS, Brazil; 5Graduate Program in Gerontology, Catholic University of Brasília, Brasília 71966-700, DF, Brazil

**Keywords:** antioxidant peptides, ROS, amphibians, neurodegenerative disorders, cancer

## Abstract

Amphibians use their skin as an effective defense mechanism against predators and microorganisms. Specialized glands produce antimicrobial peptides (AMPs) that possess antioxidant properties, effectively reducing reactive oxygen species (ROS) levels. These peptides are promising candidates for treating diseases associated with oxidative stress (OS) and redox imbalance, including neurodegenerative disorders such as Alzheimer’s disease (AD), Parkinson’s disease (PD), Huntington’s disease (HD), and amyotrophic lateral sclerosis (ALS), as well as age-related conditions, like cardiovascular diseases and cancer. This review highlights the multifaceted roles of AMPs and antioxidant peptides (AOPs) in amphibians, emphasizing their protective capabilities against oxidative damage. They scavenge ROS, activate antioxidant enzyme systems, and inhibit cellular damage. AOPs often share structural characteristics with AMPs, suggesting a potential evolutionary connection and similar biosynthetic pathways. Peptides such as brevinin-1FL and Cath-KP demonstrate neuroprotective effects, indicating their therapeutic potential in managing oxidative stress-related diseases. The antioxidant properties of amphibian-derived peptides pave the way for novel therapeutic developments. However, a deeper understanding of the molecular mechanisms underlying these peptides and their interactions with oxidative stress is essential to addressing ROS-related diseases and advancing therapeutic strategies in clinical practice.

## 1. Introduction

Amphibians have developed a wide range of peptide compounds in their skin as responses to the survival conditions, they experience, such as adaptation to the environment, the capture of their prey, and defense against their predators [1,2]. Studies have demonstrated that during their metamorphosis from egg to tadpole and ultimately to adult form, amphibians synthesize peptides in response to changes in environmental conditions within their habitat [3,4]. The transition to adulthood involves a shift from an aquatic to a terrestrial environment, presenting more significant challenges [5], such as elevated concentrations of O_2_ in the air, extreme temperature variations, and the absence of a protective barrier against ultraviolet (UV) radiation, which is typically provided by the water column [4,6]. This transition implies that amphibians must contend with a more oxidative environment, resulting in an increased production of reactive oxygen species (ROS) in response to these conditions [7]. Amphibians exhibit a critical response to oxidative stress-related damage through three primary defense mechanisms: (1) the production of antioxidant enzymes, including catalase (CAT), glutathione peroxidase (GPXs), superoxide dismutase (SOD), and peroxiredoxins, which primarily regulate endogenous antioxidant functions [8]; (2) low-molecular-weight antioxidants (LMWAs) that are not genetically encoded, such as NADH, lipoic acid, and glutathione (GSH) [9,10]; and (3) the development of antioxidant peptides (AOPs), which are typically produced through the activation of endogenous antioxidant defense systems [11] and act as direct scavengers of free radicals, similar to LMWAs [9]. The presence of AOPs in amphibian skin secretions underscores their role in regulating ROS levels [12]. This suggests that amphibians may possess AOPs that confer tolerance to unpredictable environmental changes, as they exhibit a complex array of innate eco-physiological, behavioral, and immunological traits [8] that effectively counterbalance the ROS–antioxidant imbalance.

The levels of ROS are regulated by endogenous antioxidants within redox metabolism, allowing ROS to exert their signaling functions in vital physiological processes without side effects [13]. However, when this equilibrium is disrupted, excess ROS can cause irreversible oxidative damage to biomolecules such as lipids, DNA, and proteins. This damage to these biomolecules leads to oxidative stress [14,15], and various studies have associated these alterations with the pathophysiology of many human diseases [13], including cancer [16,17]; diabetes mellitus (DM) [18]; cardiovascular diseases [19]; and neurodegenerative diseases (NDs) [13,20], including Alzheimer’s disease (AD) [21], Parkinson’s disease (PD) [22], Huntington’s disease (HD) [23], and amyotrophic lateral sclerosis (ALS) [24], in addition to accelerating physiological processes, such as aging [25,26,27]. Considering the mechanisms of action exerted by AOPs, they can be associated with effective treatments for various conditions related to oxidative stress [8], also including antifungal, antiviral, antiparasitic, anticancer, and anti-inflammatory properties, among others [28,29].

The identification of antimicrobial peptides (AMPs) in amphibian skin has been a primary focus of research due to their roles in the innate immune response, suggesting they have potential as alternatives to traditional antibiotics [28]. Various studies have evaluated additional characteristics attributed to these AMPs beyond their antimicrobial properties [12,28,30]. In this context, some AMPs may function as AOPs by chelating metals and inhibiting oxidative processes, thereby offering beneficial effects and serving as multifunctional molecules of interest for treating diseases related to oxidative stress. However, the structural diversity of AOP and AMP groups in amphibians is quite remarkable, and it could be said that practically no species has peptides with an amino acid sequence identical to that found in another. Therefore, a considerable amount of information remains unavailable, and further studies are urgently needed to understand the structure and function of these peptide groups in amphibians, as research on their molecular mechanisms in vivo remains scarce [31].

## 2. AMPs from Amphibians with Antioxidant Properties

AOPs of amphibian origin currently total more than 100 [12,30], in addition to the more than 1900 AMPs reported in the skin of amphibians for 178 species belonging to 28 genera [4]. Antimicrobial properties have been vital in the search for peptides with significant antioxidant potential, based on the ability of many amphibian species to express peptides with defensive capacity [28] and protective capacity against oxidation [8,32] to regulate microbial infections and oxidative stress, respectively. Finding antioxidant properties in AMPs has been linked to other biological properties associated with free radical scavenging capacity [29]. In addition, there is evidence of rapid action kinetics of radical elimination in amphibians, which do everything possible to protect their skin from infections or oxidative stress. Unlike AMPs, which act solely as defenses against biological injuries, AOPs can participate in both biological and non-biological injuries, leading us to conclude that defensive peptides likely act in conjunction with oxidative and microbial defenses [8].

In addition, the relationship between AOPs and AMPs in amphibians reveals a fascinating evolutionary connection. BLAST analysis in the *N*-terminal preproregion of the primary structures of different AOPs from *Rana pleuraden* specimens identified a dibasic enzymatic processing site insertion (-Lys-Arg- or -Arg-Arg-) between the spacer and mature peptide, which is also found in defensive AMP precursors from amphibians. This particularity suggests that these AOP and AMP precursors may share a similar biosynthesis pathway. Interestingly, antimicrobial assays performed on the eleven groups of AOPs from *R. pleuraden* revealed that seven can be classified as AMPs based on their antimicrobial capabilities [8]. This enables their classification as bifunctional or multifunctional peptides essential for the innate defense of amphibians [33].

This intriguing relationship between AOPs and AMPs highlights their structural similarities and paves the way for further investigation into the antioxidant properties of AMPs derived from amphibian skin secretions. Such exploration is vital for generating information relevant to developing new antioxidant formulations and potential treatments for oxidative stress-related diseases [4]. To date, the peptides with a high antioxidant potential that have been identified as being of amphibian origin have been evaluated through different in vitro [12,30,34] and in vivo methods [35], demonstrating a significant effect on diseases directly related to biomolecular damage caused by free radicals, such as premature aging [27], cancer [16], and neurological and cardiovascular diseases [36,37].

Considering that AOPs can be generated through natural production, artificial synthesis, and protein degradation, various advantages and disadvantages arise regarding their application in medicine. Among the benefits, amphibians are an excellent source of AMPs with antioxidant potential, making them noteworthy for future discoveries in the medical field. Furthermore, most of the AOPs identified and characterized from amphibians are predominantly peptides derived from research focusing on their genetic backgrounds, mechanisms, and potential applications, unlike other AOPs generated through non-natural processes such as enzymatic hydrolysis or artificial processing [3,4]. Many studies have also indicated that modification, combination therapies, and optimized administration can enhance the function and efficacy of AOPs. Structural remodeling and changes to the peptide sequences of AOPs can improve their antioxidant capabilities and other physiological functions, such as antibacterial activity and collagen synthesis, facilitating the development of multifunctional drugs [12].

Otherwise, some disadvantages should be noted. For example, most research on amphibian-derived AOPs primarily emphasizes their free radical scavenging functions, while their broader medicinal value, applications, and underlying mechanisms remain underexplored. Therefore, further progress is needed in discovering and studying amphibian-derived AOPs [3,4]. Although enzymatic hydrolysis has been applied to AOPs in frog muscle proteins, peptides produced through protein degradation tend to be unstable. Some studies have found that, despite exhibiting high antioxidant activity in vitro, certain AOPs may induce oxidative stress in vivo due to their complex mechanisms of action involving various cytokines and pathways. Consequently, clinical trials are necessary to substantiate the therapeutic potential of AOPs and to develop standardized, long-term evaluation methods for assessing their effects [12].

The antioxidant capacity of any substance, whether peptide or not, can be mediated through several mechanisms: (1) ROS scavenging capacity, (2) reducing capacity, (3) metal chelating capacity, (4) protection of biomarkers against oxidation, and (5) association with antioxidant enzymes [38]. Among the methods frequently used to determine the antioxidant capacity are in vitro methods that measure a peptide’s ability to scavenge free radicals, such as DPPH and ABTS [8], which are the most common due to their relative stability, easy measurement, and good reproducibility [39], as well as the oxygen radical absorbance capacity (ORAC) and the reducing power in the transformation of Fe^3+^ to Fe^2+^ [40]. All these methods have made it possible to determine the antioxidant capacity of peptides identified in different amphibian species, preferably by determining the DPPH and ABTS^•+^ radical scavenging capacity.

As mentioned, other antioxidant assays used to determine AOPs derived from amphibians include in vitro and in vivo methods based on measuring redox enzymes, offering insights into the degree of oxidative damage to cells [41]. Enzymes such as CAT, SOD, and lactate dehydrogenase (LDH), along with mediators like GSH and even the ROS themselves, can be detected using colorimetric and fluorometric assays from treated cell lines. Depending on the methodology and kits employed, these assays can target culture supernatants or cellular lysates [42]. For in vivo experiments, these assays are performed by extracting tissues post-treatment, which, in conjunction with histological analysis, provides information about cellular damage [11]. Additionally, specific genes related to inflammatory processes during treatment can be monitored by qRT-PCR [43], aiding in the clinical determination of pathological states. Another method involves detecting cytotoxic oxidative stress biomarkers, like malondialdehyde (MDA) and 4-hydroxynonenal, modified molecules or reaction products generated by ROS in the surrounding environment [44,45]. It is essential to highlight that antioxidant mechanisms and the results evaluated by different antioxidant tests can vary widely; therefore, comparing one method with another is difficult. Considering the wide variety of structural characteristics of AOPs, antioxidant tests should not be restricted to a single model. Thus, it is advisable to use different methods to evaluate their antioxidant properties and more accurately indicate their possible protective effects [12].

However, the effectiveness of AOPs has been attributed to the participation of amino acids with chemical properties in their side chains that are responsible for reducibility, hydrophobicity, and electron transfer capacity, promoting interaction with free radicals and neutralization through antioxidation chemical reactions [46,47]. In general, the more reducing residues there are in the sequence, the stronger the antioxidant properties of the AOPs will be [46]. For example, aromatic amino acids such as Phe [48,49], Tyr [50], and Trp [20], common in the structures of AMPs and AOPs, have contributed significantly to amphibian-derived peptide’s antioxidant properties. Previous studies have shown that these aromatic residues can act as electron donors due to their electron-rich aromatic structure, and they may help stabilize ROS through interactions that involve direct electron transfer [47]. It has also been reported that Tyr and Trp are highly effective in protecting cells because the transmembrane domains of integral membrane proteins exhibit substantial enrichment of both residues, widespread in regions of higher lipid density [51]. These residues exert essential antioxidant functions within the lipid bilayer and protect cells from oxidative damage, acting as inhibitors of lipid peroxidation and oxidative cell death [52], due to their relatively stable phenoxy and indolyl radicals, which can capture free radicals, thereby achieving the function of scavenging free radicals [53]. Histidine (His) exhibits strong antioxidant properties due to its imidazole rings acting as proton donors. In addition to His, Lys and Val can scavenge hydroxyl radicals by serving as electron donors, creating a hydrophobic microenvironment within the molecule that enhances the peptide’s antioxidant capacity [54].

Sulfur-containing amino acids such as Met and Cys have also been suggested to be important in the antioxidant function of peptides [20,50], since the unpaired thiol group of Cys is crucial for free radical scavenging, attributed to its ability to donate an electron to a radical [55]. Another study indicates that AOPs with a free Cys are the dominant components of highly antioxidant peptides in the species *Odorrana andersonii* [56], where the scavenging activity of such peptides would be mainly related to the disulfide bridge and, secondarily, to the hydroxyl groups present in the synthesized peptide sequence, thus contributing to their hydrogen donation capacity [57]. Additionally, other studies confirm that the antioxidant activity of some AOPs from the pleurain, odorranain, and brevinin families directly correlates with the number of Cys residues and disulfide bridges [58]. For instance, it was demonstrated that the antioxidant activity of cyclic peptides with Cys residues forming a disulfide bridge was lower than that of linear peptides. This finding might imply that linear peptides with free Cys residues play a more critical role in the skin’s antioxidant defense system [59].

This study gathers research on AMPs with antioxidant properties, focusing on their amino acid content across various amphibian families, especially in the *Ranidae* family [28,30], as listed in Table 1. These peptide families have demonstrated important physicochemical properties linked to mechanisms that protect the skin from free radicals, suggesting promising potential for other pharmacological applications [28,29,60,61,62,63].

### 2.1. Antioxidin Family

As the first peptide group with antioxidant potential reported, the antioxidins have demonstrated their capacity to eliminate free radicals in different studies, leading to ongoing research on this peptide family to this day [35]. The first AOPs to be discovered were antioxidin-RP1 (AMRLTYNKPCLYGT) and antioxidin-RP2 (SMRLTYNKPCLYGT), 14-amino acid peptides derived from the *Rana pleuraden* species. Antioxidin-RP1 was shown to be the most potent peptide in scavenging free radicals of DPPH and ABTS^•+^, capable of eliminating 100% of the radicals at a concentration of 80 μg·mL^−1^. It exhibited the highest capacity to reduce Fe^3+^ ions to Fe^2+^ compared to other AOPs and the highest release of nitric oxide (NO), produced by the sodium nitroprusside dihydrate (SNP) NO donor [8].

Another AOP belonging to this class is antioxidin-RL (AMRLTYNRPCIYAT), obtained from *Odorrana livida*, which was characterized by its potent free radical scavenging capacity, determined through ABTS^•+^ assays [64] and strong in vitro and in vivo UVB-induced response tests [42]. Specifically, cytotoxicity assessed using the MTT assay indicated that the viability of human keratinocyte (HaCaT) and human skin fibroblast (HSF) cells treated with antioxidin-RL in the absence of UVB irradiation was comparable to that of untreated control cells. Flow cytometry experiments employing DCFH-DA (2′,7′-dichlorodihydrofluorescein diacetate), a fluorescent redox probe for detecting ROS, revealed that antioxidin-RL provided a superior protective effect on UVB-irradiated HaCaT cells compared to vitamin C at all tested concentrations (5, 10, 20, 50, and 100 μg·mL^−1^). A similar effect was observed in irradiated HSF cells when the peptide concentration exceeded 20 μg·mL^−1^. Treatment with 50 μg·mL^−1^ of antioxidin-RL or vitamin C after UVB irradiation resulted in a reduction of ROS production by 45.44% and 49.22% in HaCaT cells, respectively, and by 52.49% and 58.24% in HSF cells, respectively. It was also observed that antioxidin-RL effectively penetrates the cell membrane and positively influences cell migration. This led to increased collagen deposition in the histological analysis of skin from mice exposed to UVB radiation, followed by treatment with the peptide. Additionally, the levels of GSH and LDH in the culture media of treated and untreated cell lines exposed to radiation were assessed using commercially available ELISA kits. UVB-irradiated HaCaT and HSF cells exhibited significantly reduced GSH levels of 36.57% and 59.6%, respectively, compared to non-irradiated cells. However, treatment with antioxidin-RL or vitamin C following UVB irradiation restored GSH levels to 85.34% and 92.68% in HaCaT cells and 68.24% and 78.82% in HSF cells, respectively. Regarding LDH release, UVB irradiation markedly stimulated LDH release by 159% and 204% in HaCaT and HSF cells, respectively. Furthermore, treatment with antioxidin-RL or vitamin C at a concentration of 50 μg·mL^−1^ after UVB irradiation decreased LDH release to 18.33% and 28.33% in HaCaT cells, and 23.19% and 28.99% in HSF cells [42]. Currently, only antioxidin-RL [8] and OA-VI12 (VIPFLACRPLGL) [11], isolated from *O. livida* and *O. andersonii*, respectively, have been described with possible protective effects on amphibian skin through in vivo assays [11,42], showing that they prevent UVB irradiation-induced photoaging in mice. However, the mechanisms of action of these two AOPs are still not fully understood [35].

Antioxidin-2 (YMRLTYNRPCIYAT) was designed as an analog of antioxidin-RL by replacing the *N*-terminal Ala with Tyr to improve free radical binding capacity. Antioxidant testing of this analog was performed using in-solution free radical scavenging assays, and the results showed that it scavenged free radicals faster than antioxidin-RL [66]. However, using PC-12 cell models (a mouse neuronal cell line-derived pheochromocytoma) demonstrated that, although antioxidin-2 and antioxidin-RL inhibited the accumulation of intracellular free radicals induced by H_2_O_2_, they preserved mitochondrial morphology and reduced the expression of dynamin-related protein-1 in mitochondria. Antioxidin-RL was shown to be more effective in preventing the dissipation of the mitochondrial membrane potential [66].

Antioxidin-I (TWYFITPYIPDK), a 12-amino acid peptide identified in *Physalaemus nattereri*, was evaluated through in vitro free radical scavenging tests and showed poor performance in most free radical scavenging assays, except for the ORAC assay. However, when tested on murine fibroblasts, antioxidin-I exhibited low cytotoxicity by suppressing menadione-induced redox imbalance. Furthermore, it substantially attenuated hypoxia-induced ROS production when tested on live microglial cells exposed to hypoxia, suggesting a possible neuroprotective role for this peptide. Antioxidin-I is found in the skin tissue of three additional tropical frog species, *Phyllomedusa tarsius*, *P. distincta*, and *Pithecopus rohdei* [48]. In another study, antioxidin-NV (GWANTLKNVAGGLCKMTGAA), obtained from the frog *Nanorana ventripunctata*, was a protective peptide against skin photodamage. It reduced skin erythema, thickness, and wrinkle formation caused by UVB exposure in hairless mice. This AOP directly and rapidly scavenged excess ROS after UVB irradiation, alleviating UVB-induced DNA damage, cell apoptosis, and inflammatory response, thereby protecting against UVB-induced skin photoaging. The antioxidant activity was determined by the ABTS method, confirming that the properties of antioxidin-NV make it a promising candidate for developing a novel antiphotoaging agent [35].

Studies identified the peptide antioxidin-PN (FLPSSPWNEGTYVLKKLKS) from the total RNA expression of *Pelophylax nigromaculatus*, also present in the same species but from different specimens distributed in two provinces of China: Yunnan and Guizhou. The peptide was evaluated for its antioxidant properties, demonstrating, like the other eight synthesized peptides, substantial elimination of ABTS^•+^ free radicals in a dose-dependent manner, eliminating 30% of the ABTS^•+^ radical at 15 s and 100% in 10 min at a low concentration (6.25 μg·mL^−1^, approximately 2.85 µM). The results of this study allowed us to determine that gene-environment interaction is an essential factor in the diversity of bioactive peptides within the same species. Nonetheless, additional research is required to further elucidate its action mechanisms [65].

One study supports the notion that the antioxidant potential of peptides such as antioxidins may be due to the content of Pro residues since all those found there presented at least one in their structure [8]. The study demonstrated that Pro protects cells against H_2_O_2_, butyl tert-hydroperoxide, and a carcinogenic inducer of oxidative stress [82].

### 2.2. Pleurain Family

Pleurain is another AMP group found in the skin of *Rana pleuraden*, where in addition to the antioxidin-RP1 and antioxidin-RP1, the identification of 13 new potential antioxidant peptide groups, comprising 36 members, was reported through a peptidomic and genomic approach [8]. In an earlier study, the antioxidant group pleurain-A had already been reported [3]. Of the total, 11 of the 14 groups exerted antioxidant activity (pleurain-A, -D, -E, -G, -J, -K, -M, -N, -P, -R, and antioxidin-RP). Some groups, such as pleurain-B and pleurain-N, exerted only antimicrobial and antioxidant activity, respectively, while others, such as pleurain-G, showed both antimicrobial and antioxidant activities. All peptide groups share the characteristic of containing Pro residues, suggesting that this amino acid may play an essential role in their antioxidant capacity. This conclusion is substantiated by the study’s emphasis on the molecular basis of AOPs. It was noted that all amino acid residues associated with antioxidant activity, including Pro, Met, Cys, Tyr, and Trp in their free forms, were substituted with Gly in the eleven peptide groups examined. The results indicated that substituting Pro, Met, free Cys, or Trp significantly diminished their antioxidant capability, while substituting Trp exerted only a marginal effect. Additionally, it was confirmed that replacing all these amino acid residues entirely abolished their antioxidant function [8].

### 2.3. Cathelicidin Family

Among the earliest groups of peptides documented in amphibians are the cathelicidins. While numerous cathelicidins have been identified in mammals [83], few have been reported in amphibians, with only eight cathelicidins having been recorded in frogs [67,68]. Recently, cathelicidin-OA1 (IGRDPTWSHLAASCLKCIFDDLPKTHN, featuring a disulfide bridge) was identified from the skin of *Odorrana andersonii*, produced by post-translational processing of a 198-residue prepropeptide [67]. Functional analysis showed that cathelicidin-OA1 did not generate direct microbial killing, acute toxicity, or hemolytic activity, but it exhibited antioxidant activity, evidenced by the radical scavenging activity of ABTS^•+^ and DPPH. Compared with other odorous frog peptides, such as adersonin-AOP1 (FLPGLECVW), the antioxidant activity of cathelicidin-OA1 was slightly weaker. However, this activity was enhanced when the intramolecular disulfide bridge of cathelicidin-OA1 was broken, resulting in linear cathelicidin-OA1 with two free Cys residues. An enhancement in the antioxidant activity was also observed in the mutant cathelicidin-OA1 (Cys14/Ala), which contains one free Cys residue due to the substitution of Cys-14 with Ala. Research supports the notion that Cys enhances the antioxidant activities of specific peptides, except for tylotoin (KCVRQNKRVCK), an amphibian cathelicidin known for its direct microbicidal effects. Additionally, cathelicidin-OA1 has been demonstrated to facilitate wound healing in HaCaT and skin HSF by promoting cell proliferation [67].

Cathelicidin-NV (ARGKKECKDDRCRLLMKRGSFSY), which was previously identified in the spot-bellied plateau frog (*Nanorana ventripunctata*), demonstrated an ability to alleviate UVB-induced skin photoaging in mice. Additionally, this peptide effectively suppressed cytotoxicity, DNA fragmentation, and apoptosis. It also decreased the protein expression levels of c-Jun N-terminal kinase (JNK), transcription factor Jun (c-Jun), and matrix metalloproteinase-1 (MMP-1), all of which play a role in regulating collagen degradation in HaCaT cells exposed to UVB irradiation. Cathelicidin-NV directly scavenged excess intracellular ROS to protect HaCaT cells, thereby alleviating photoaging and positioning it as an excellent candidate for preventing and treating UV-induced skin photoaging [68].

Additionally, a novel peptide from the cathelicidin family, named Cath-KP (GCSGRFCNLFNNRRPGRLTLIHRPGGDKRTSTGLIYV), was identified from the skin of the Asian painted frog, *Kaloula pulchra*. Circular dichroism and homology modeling indicated an α-helix conformation for Cath-KP. The results demonstrated antioxidant properties through free radical scavenging and iron reduction analyses. The administration of Cath-KP in a dopamine-induced cell line and in PD mice, induced by 1-methyl-4-phenyl-1,2,3,6-tetrahydropyridine (MPTP)—a precursor to the monoaminergic neurotoxin 1-methyl-4-phenylpyridinium (MPP^+^), which causes permanent symptoms of PD by destroying dopaminergic neurons—revealed its ability to penetrate cells and reach deep brain tissues. This resulted in increased cell viability and reduced oxidative stress-induced damage by promoting the expression of antioxidant enzymes [69].

### 2.4. Spinosan Family

In the investigation of *Paa spinosa* skin, precursor cDNAs of four new AMPs were cloned, and subsequent peptide sequencing revealed the existence of a new family of frog AMPs distinct from those previously reported in other amphibians. The four identified peptides were spinosan-A (DLGKASYPIAYS), spinosan-B (DYCKPEECDYYFSFPI), spinosan-C (DLSMMRKAGSNIVCGLNGLC), and spinosan-D (MEELYKEIDDCVNYGNCKTLKLM). These peptides were chemically synthesized and evaluated for their antimicrobial, antioxidant, hemolytic, and cytotoxic activities [70]. The peptides demonstrated a weak hemolytic effect against rabbit erythrocytes while exhibiting a robust antioxidant effect through free radical scavenging methods. In the DPPH free radical scavenging assay, spinosan-A, -B, -C, and -D demonstrated significant radical scavenging activity at a concentration of 80 μg·mL^−1^. Notably, spinosan-C achieved a free radical scavenging percentage of 85%, whereas spinosan-A, -B, and -D exhibited respective scavenging values of 52%, 59%, and 49%. The study posits that peptides with potential antioxidant activity invariably contain residues such as Cys, Pro, Met, Tyr, or Trp, which are responsible for free radical scavenging activity [70].

### 2.5. Jindongenin and Palustrin Families

Jindongenin-1a (DSMGAVKLAKLLIDKMKCEVTKAC) is a 24-amino acid peptide belonging to the jindongenin AMP peptide family, reported from the Chinese torrent frog *Amolops jingdongensis*. The palustrin family has also been identified within the same species, comprising palustrin-2AJ1 (GFMDTAKNVAKNVAVTLIDKLRCKVTGGC) and palustrin-2AJ2 (GFMDTAKQVAKNVAVTLIDKLRCKVTGGC), both of which exhibit structural similarities to jindongenin-1a. These three peptides share high sequence similarity within their signal and propeptide regions (42 amino acids). Such similarities suggest that these two families of AMPs may have evolved from an ancestral gene through mechanisms such as genetic duplication, splicing, or domain shuffling. Jindongenin-1a and palustrin-2AJ1 were synthesized to evaluate their antimicrobial, hemolytic, antioxidant, and cytotoxic activities. Both peptides demonstrated broad-spectrum antimicrobial activity against standard and clinically isolated bacterial strains. However, DPPH free radical scavenging assays revealed that jindongenin-1a and palustrin-2AJ1 exhibited low antioxidant activity, with inhibition rates of 8% at 32 mM and 6% at 25 mM, respectively. These findings suggest that these peptides do not play a significant role in adapting *A. jingdongensis* to intense UV exposure; future studies may clarify the factors contributing to this adaptation [34].

However, a study conducted on three East Asian frog species—*Amolops lifanensis*, *Amolops granulosus*, and *Hylarana taipehensis*—identified a peptide group termed palustrin-2GN from *A. granulosus*, alongside other essential groups of AOPs and AMPs, such as temporin and brevinine. The peptides palustrin-2GN1 (GLWNTIKEAGKKFALNLLDKIRCGIAGGCKG), palustrin-2GN2 (GFMDTAKNVFKNVAVTLLDKLKCKIAGGC), and palustrin-2GN3 (GILD-TLKQLGKAAAQSLLSKAACKLAKTC) exhibited antimicrobial activity against various Gram-positive strains, including *Staphylococcus aureus* (ATCC 25923) and *Enterococcus faecium* 091299 (IS), as well as Gram-negative strains, such as *Pseudomonas aeruginosa* (CGMCC 1.50), *Klebsiella pneumoniae* 08040724 (IS), *Enterobacter cloacae* (CGMCC 1.57), and *Escherichia coli* (ATCC 25922), among others. Notably, only palustrin-2GN1 demonstrated a weak effect on scavenging free radicals ABTS^•+^ and DPPH [32]. This research and prior studies suggest that amino acids such as Cys and others play a significant role in the antioxidant capabilities of peptides [8,50]. While the structures of these AMPs include one or more of these amino acids, the findings also suggest that peptides containing these residues do not necessarily exhibit antioxidant activity. For instance, palustrin-2GN2 displayed no antioxidant activity in the free radical scavenging assay despite containing one or more amino acids [32].

### 2.6. Taipehensin Family

From *Hylarana taipehensis*, a frog from East Asia, two AOPs denominated taipehensin-1TP1 (TLIWEFYHQILDEYNKENKG) and taipehensin-2TP1 (CLMARPNYRCKIFKQC) were identified. These AOPs exhibited a strong ABTS^•+^ free radical scavenging capacity and efficient DPPH free radical scavenging kinetic activity, similar to other reported AOPs [8,64]. The free radical scavenging rates of ABTS^•+^ and DPPH were determined after 30 min. At a concentration of 200 mM, taipehensin-1TP1 rapidly scavenged 34.1% of ABTS^•+^ in 1 min, with a final scavenging rate of 65.4% after 30 min. Among these peptides, taipehensin-2TP1 (>50 mM) showed the highest scavenging ability and scavenged almost all ABTS^•+^ in 1 min. The ability of taipehensin-2TP1 to scavenge DPPH free radicals was lower than its ability to scavenge ABTS^•+^. At 50 mM, taipehensin-2TP1 scavenged only 85.4% of DPPH free radicals in 30 min. At a concentration of 200 mM, it showed scavenging solid ability for ABTS^•+^ and DPPH free radicals, with scavenging rates of 99.8% and 98.8%, respectively [32].

### 2.7. Brevinin Family

Brevinin-1TP1 (FLPGLIKAAVGVGSTILCKITKKC), brevinin-1TP2 (FLPGLIKAAVGIGSTIFCKISKKC), brevinin-1TP3 (FLPGLIKVAVGVGSTILCKITKKC), brevinin-1TP4 (FLPGLIKAAVGIGSTIFCKISRKC), brevinin-2TP1 (SILSTLKDVGISAIKSAGSGVLSTLLCKLNKNC), and brevinin-2TP2 (SILSTLKDVGISALKNAGSGVLKTLLCKLNKNCEK) from *Hylarana taipehensis*, as well as brevinin-1LF1 (FLPMLAGLAANFLPKIICKITKKC), brevinin-1LF2 (FLPIVASLAANFLPKIICKITKKC), brevinin-2LF1 (GFMDTAKNVAKNVAKNVAVTLLDKLRCKVTGGC), and brevinin-2LF2 (SIMSTLKQFGISAIKGAAQNVLGVLSCKIAKTC) from *Amolops lifanensis*, are examples belonging to this family. Some of these AMPs, as is the case with brevinin-1TP1, brevinin-1TP2, brevinin-1TP3, and brevinin-1LF, are active against a broad microbial spectrum, as well as having ABTS^•+^ or DPPH free radical scavenging capacity. Others, such as brevinin-2LF1, presented an MIC of 3.1 µM against the Gram-negative bacterium *Psychrobacter faecalis* X29. Additionally, brevinin-1TP1 showed both antioxidant and antimicrobial activity, with a free radical scavenging percentage of ABTS^•+^ of 26.5% at a concentration of 200 µM and suitable minimum inhibitory concentrations against different microbial strains, such as *Staphylococcus aureus* (ATCC 25923) with MIC = 12.5 µM, *Enterococcus faecalis* 981 with MIC = 6.3 µM, *Nocardia asteroides* 201118 (IS) with MIC = 6.3 µM, *Psychrobacter faecalis* X29 with MIC = 6.3 µM, and *Candida glabrata* 090902 with MIC = 12.5 µM [32].

Another AOP identified in this family is brevinin-1FL (FWERCSRWLLN), derived from the skin of the frog *Fejervarya limnocharis*. Functional analysis demonstrated that brevinin-1FL could scavenge ABTS^•+^, DPPH, NO, and hydroxyl radicals in a concentration-dependent manner while reducing iron oxidation. Furthermore, brevinin-1FL was found to display neuroprotective activity by reducing MDA and ROS levels, enhancing the activity of endogenous antioxidant enzymes, and inhibiting H_2_O_2_-induced death, apoptosis, and cell cycle arrest in PC-12 cells, which was associated with its regulation of the AKT/MAPK/NF-kB signaling pathways. Furthermore, brevinin-1FL decreased paw edema and lowered levels of TNF-alpha, IL-1β, IL-6, myeloperoxidase (MPO), and MDA, while restoring CAT and SOD activity, as well as GSH content in carrageenan-injected mice. The findings suggest that brevinin-1FL has significant therapeutic potential for diseases associated with oxidative damage [71].

### 2.8. Nigroain Family

In other studies, 50 peptides classified into 21 peptide families with antioxidant or antimicrobial activity were identified from the species *Amolops daiyunensis*, *Hylarana maosuoensis*, *Pelophylax hubeiensis*, and *Nanorana pleskei*, belonging to the *Ranidae* and *Dicroglossidae* families. From *H. maosuoensis*, peptides belonging to the nigroain family were identified, such as nigroain-B-MS1 (CVVSSGWKWNYKIRCKLTGNC), nigroain-C-MS1 (FKTWKNRPILSSCSGIIKG), nigroain-D-SN1 (CQWQFISPSRAGCIGP), and nigroain-K-SN1 (SLWETIKNAGKGFILNILDKIRCKVAGGCKT). Antioxidant and antimicrobial activity assays showed that some of these peptides exhibited significant results. For example, at a concentration of 50 μM, both nigroain-B-MS1 and nigroain-C-MS1 showed relatively strong ABTS^•+^ and DPPH free radical scavenging abilities, with scavenging rates of 99.7% and 68.3% for nigroain-B-MS1, and 99.8% and 58.3% for nigroain-C-MS1, respectively. Nigroain-B-MS1 also showed activity against Gram-positive bacteria such as *Staphylococcus aureus* (ATCC 25923) with MIC = 4.7 μM and *Nocardia asteroides* 201118 (IS) with MIC = 18.8 μM. Nigroain-C-MS1 only showed antimicrobial activity against the same strain of *Nocardia* with a MIC of 150 μM. Interestingly, these two peptides showed weak hemolytic activity [72].

### 2.9. Andersonin Family

From *Odorrana andersonii*, a Chinese odorous frog, 38 novel mature AMPs grouped into 25 different subfamilies were identified: andersonin-A, -B, -C, -E, -F, -J, -L, -M, -N, -O, -P, -R, -S, -T, -U, and -X (all with only one peptide); andersonin-I, -K, -Q, -V, and -Y (all with two peptides); and andersonin-D, -G, -H, and -W (all with three peptides). Eight peptides were synthesized and assayed for antimicrobial, antioxidant, and other activities. Only andersonin-C1 (TSRCIFYRRKKCS) and andersonin-D1 (FIFPKKNIINSLFGR) showed killing effects against the tested strains. Andersonin-C1 showed MICs of 30 μg·mL^−1^ against *E. coli*, *C. albicans*, as well as *B. pyocyaneus*, and MIC > 120 μg·mL^−1^ against *S. aureus*. Meanwhile, andersonin-D1 showed MIC > 126 μg·mL^−1^ against *E. coli*, but MICs of 16 μg·mL^−1^ against *C. albicans*, *B. pyocyaneus*, and *S. aureus*. Andersonin-C1, andersonin-N1 (ENMFNIKSSVESDSFWG), and andersonin-R1 (ENAEEDIVLMENLFCSYIVGSADSFWT) showed a scavenging rate percentage against ABTS^•+^ radicals greater than 70%. Interestingly, andersonin-C1 features a cyclic motif with a disulfide bond at the *C*-terminus, showing potent antioxidant activity. However, others, such as andersonin-G1 (KEKLKLKCKAPKCYNDKLACT) and andersonin-H3 (VAIYGRDDRSDVCRQVQHNWLVCDTY), both with a cyclic motif as well, showed lower antioxidant activity, with scavenging rate percentages against ABTS^•+^ radicals greater than 20% and 30%, respectively. Only the peptides andersonin-D1, andersonin-Q1 (EMLKKKKEVKMERKT), and andersonin-S1 (DANVENGEDAEDLTDKFIGLMG) did not show detectable antioxidant activity [59]. The antioxidant activity of cyclic peptides was lower than that of linear peptides when compared with other families of AMPs with free Cys. Some studies support the proposition that peptides with potential antioxidant activity always contain a free Cys residue responsible for the free radical scavenging activity [64].

Considering that most studies on amphibians have been limited to the genetic mechanisms of adaptation to low oxygen levels and temperature in high-altitude areas, it was found that very few studies focused on the adaptation of amphibians to UV radiation according to altitude, thus demonstrating the development of specialized molecules in response to such conditions. Therefore, in a study conducted on two species of frogs present in different altitudinal zones—specifically, the frog *Odorrana andersonii*, found in plateau areas (approximately 2500 m) of Yunnan province in China, which is subject to long exposures to sunlight and intense UV radiation; and the cave frog *O. wuchuanensis*, distributed in a small number of caves that do not receive light throughout its life cycle at an altitude of 800 m in Guizhou province—it was found that the peptide families of these two species differ significantly in their antioxidant potential. *O. andersonii* exhibited peptides with greater diversity and free radical scavenging potential against UV radiation than those in *O. wuchuanensis.* Here, 26 new subfamilies belonging to the andersonin family were identified from high-altitude areas for *O. andersonii.* Most of these AOP subfamilies contained only one member, but others exhibited high diversity, such as andersonin-AOP8 and andersonin-AOP14, which included six and five members, respectively. Andersonin-AOP1 (FLPCLECVN) was the shortest AOP, while andersonin-AOP25 (ATALGIPPRGFLPIVNKFKDIILC) and andersonin-AOP26 (IPWKLPATLRPVENPFSKPLCRNY) were the longest AOPs (24 amino acids each). The antioxidant activity results for AOPs showed that AOPs from *O. andersonii* skin exhibited ABTS^•+^ scavenging activities more significant than 90% to 50 μM, except for andersonin-AOP9 (LKGFEMGMDMKRT, 51.33%), andersonin-AOP13 (APDRPRKFCGILG, 84.69%), andersonin-AOP17 (VTPPWARIYYGCAKA, 66.67%), and andersonin-AOP19a (GAGFWKMGKYGQKRRD, 60.00%). These results were further confirmed by determining DPPH free radical scavenging activity, as most of the AOPs from *O. andersonii* completely scavenged DPPH radicals, suggesting that *O. andersonii* has developed a much more complex and compelling skin AOP system to survive high-altitude UV radiation levels [56].

### 2.10. Odorranain Family

Odorrana has the most abundant and diversified AMPs among all studied amphibian genera. Even from a single frog, 46 cDNA sequences encoding precursors of 22 different AMPs were characterized from the skin of *Odorrana tiannanensis*. Ten peptides, grouped into six subfamilies, correspond to the odorranain family. Specifically, two AMPs, odorranain-C7HSa (SLLGTVKDLLIGAGKSAAQSVLKGLSCKLSKDC) and odorranain-G-OT (FVPAILCSILKTC), were purified from the skin secretions of *O. tiannanensis*, and their amino acid sequences matched the deduced sequences of the cDNAs. Odorranain-A-OT (VVKCSFRPGSPAPRCK), identified from the cDNA sequences, has the most potent antioxidant activity, scavenging 89.24% of the DPPH radicals, followed by odorranain-G-OT, which exhibited a scavenging rate of 70.41% at a concentration of 80 mg·mL^−1^. The study argues that the scavenging activity of the peptides is mainly related to the disulfide bridge and the hydroxyl groups from the main chain of residues such as Ser and Thr, which possibly contribute to their hydrogen-donating capacity [58].

### 2.11. Hainanenin Family

From the skin secretions of the frog *Amolops hainanensis*, two new AMPs belonging to this family were identified from the cloning of 31 cDNA sequences encoding ten new AMPs across four peptide families. These two new peptides are hainanenin-1 (FALGAVTKLLPSLLCMITRKC) and hainanenin-5 (FALGAVTKRLPSLFCLITRKC), each consisting of 21 amino acid residues with a seven-residue *C*-terminal disulfide loop between Cys-15 and Cys-21. Hainanenin-1 and -5 were synthesized and tested in vitro for antimicrobial, antioxidant, and hemolytic activities. The results showed that both AMPs possessed robust, broad-spectrum antimicrobial activities against Gram-positive and Gram-negative bacteria and fungi, including many clinically isolated drug-resistant pathogenic microorganisms. However, they exhibited slight antioxidant activity and undesirable hemolytic responses in human erythrocytes [55]. Although one aim of the study was to demonstrate that most of the peptides display multiple functions, such as antioxidant and antimicrobial activities [8,33,64], it was observed that at concentrations up to 160 μg·mL^−1^, hainanenin-1 and -5 showed no apparent antioxidant activity against DPPH free radicals, with free radical scavenging percentages of 3.60% and 1.52%, respectively. The thiol groups of cysteines are crucial for the free radical scavenging activity of peptides, as they can donate electrons to a free radical [50,64]. However, it is concluded that the thiol groups of the two cysteines in hainanenin-1 and -5 are oxidized and form a disulfide bridge, which may cause the absence of their antioxidant activities [55].

### 2.12. FW Family

Another peptide family is FW, found in the tree frog *Hyla annectansis*, inhabiting the southwestern plateau area of China, where there is intense UV radiation and long periods of sunlight. This indicates that its bare skin may contain chemical defense components that protect it from acute photodamage [43]. Despite extensive research, no peptide demonstrating this effect has been identified. In this study, however, two novel peptides, FW-1 (FWPLI(NH_2_)) and FW-2 (FWPMI(NH_2_)), were discovered, exhibiting potential antioxidant effects in the epidermis by reducing UVB-induced ROS production through an unknown mechanism. The impact of FW-1 and FW-2 on UVB-induced ROS accumulation were tested using the fluorometric assay that uses DCFH-DA as a reagent. This reveals that ROS production significantly increased when HaCat cells were irradiated with UVB. Importantly, with the pretreatment of HaCet cell with both peptides for 1 h, a significantly attenuated UVB-induced ROS production was observed, demonstrating remarkable inhibition at a concentration of 12 mg·mL^−1^—comparable to the positive control, *N*-acetylcysteine (NAC). Additionally, both HaCaT cells and the supernatant were analyzed using qRT-PCR and ELISA to examine the mRNA and protein levels of the pro-inflammatory cytokines TNF-α and IL-6. Protein levels of both TNF-α and IL-6 were notably increased in the skin of mice exposed to UVB radiation. However, treatment with the peptides FW-1 and FW-2 significantly inhibited the secretion of these cytokines compared to the vehicle-treated control group. In vivo experiments with depilated mice, intradermally injected with 5 mg·kg^−1^ of FW-1 and FW-2 and subjected to a single dose of UVB irradiation for 48 h, demonstrated that UVB exposure caused burns, edema, blisters, and peeling. Mice treated with FW-1 and FW-2 exhibited less skin damage and better symptom relief. Histological sections and hematoxylin and eosin staining revealed significantly reduced skin swelling and immune cell infiltration in UVB-irradiated skin in treated mice compared to controls. Given their ease of production, storage, and potential photoprotective activity, FW-1 and FW-2 could serve as promising lead compounds for developing new pharmacological agents to suppress UVB-induced skin inflammation. Additionally, this study enhances our understanding of the defensive mechanisms of tree frog skin against UVB irradiation [43].

### 2.13. OM Family

From the OM family, obtained from skin secretions of odorous frogs *Odorrana margaretae*, a novel peptide was identified, produced by post-translational processing of a 71-residue prepropeptide. This peptide, OM-LV20 (LVGKLLKGAVGDVCGLLPIC), contains an intramolecular disulfide bridge at the *C*-terminus and exhibits weak antioxidant activity. Although it had no direct antimicrobial effects, hemolytic activity, or acute toxicity, OM-LV20 effectively promoted wound healing in HaCaT and HSF in both a time- and dose-dependent manner. For that, OM-LV20 provides a new template for bioactive peptides in developing novel wound healing agents and drugs [74].

A novel AOP produced by post-translational processing of a 61-residue prepropeptide was also discovered from the skin secretions of *O. margaretae* and was named OM-GF17 (GFFKWHPRCGEEHSMWT). OM-GF17 did not exhibit direct antimicrobial activity but could scavenge free radicals like ABTS^•+^, DPPH, NO and reduce oxidized Fe^3+^ ions. However, these activities were slightly weaker than other peptides identified in the odorous frog, such as adersonin-AOP1 [56]. The results also demonstrated that five amino acid residues, namely Cys, Pro, Met, Trp, and Phe, are related to the antioxidant activity of OM-GF17. There was no apparent influence on ABTS^•+^ radical scavenging activity when all five amino acids were mutated individually; however, the antioxidant activity was abolished when all five amino acids were mutated together. When Met-15 and Trp-16 were replaced, the new mutant showed a decrease in scavenging rate to approximately 50 s, and when Trp-5 and Cys-9 were replaced, the scavenging rate decreased to approximately 60 s. OM-GF17 reached a maximum scavenging rate in 10 s. These five amino acids influence ABTS^•+^ scavenging efficiency; specifically, Phe-2, Phe-3, and Pro-7 showed greater responsibility for the scavenging efficiency of OM-GF17. Additionally, it was observed that the Cys-9 mutant of OM-GF17 resulted in eliminated Fe^3+^-reducing power, while for (Phe-2/Ala), (Phe-3/Ala), (Trp-5/Ala), (Pro-7/Ala), (Met-15/Ala), and (Trp-16/Ala), mutants of OM-GF17 showed similar activity to the natural parental peptide. The fact that Cys-9 is responsible for the peptide’s Fe^3+^-reducing power is significant. As a result, the antioxidant activity of amphibian peptides may be closely linked to their amino acid composition and sequence. Although these five residues are generally recognized as contributing to the antioxidant activity of OM-GF17, this does not imply that all of them directly participate in free radical quenching. Furthermore, this novel gene-encoded antioxidant peptide could facilitate the development of new antioxidant agents [73].

In another study, a novel antioxidant peptide named OM-GL15 (GLLSGHYGRASPVAC) was identified from the skin of the green odorous frog *O. margaretae*. Its antioxidant activity was demonstrated by evaluating its ability to scavenge ABTS^•+^ and DPPH free radicals and its power to reduce Fe^3+^ to Fe^2+^. Exploration of the underlying mechanisms further demonstrated that OM-GL15 exerts significant antioxidant potential by reducing lipid peroxidation and MDA levels, protecting epidermal cells from UVB-induced apoptosis. It inhibits DNA damage by downregulating p53, caspase-3, caspase-9, and Bax and upregulating Bcl-2. Additionally, the topical application of OM-GL15 significantly reduced UVB-induced skin photodamage in mice. These results emphasize the potential use of amphibian skin-derived peptides for protection against UVB-induced photodamage and present a new candidate peptide for developing anti-photodamage agents [31].

### 2.14. OA Family

In a previous study of the skin secretions of the frog *O. andersonii*, coding a short gene led to peptide OA-VI12 (VIPFLACRPLGL) expression. This peptide was shown to exert a direct scavenging capacity for free radicals, suggesting a possible role in protecting the skin from photodamage due to its high-altitude habitat [67]. Since OA-VI12 preserved cell viability in immortalized HaCaT, it was analyzed in models of oxidative stress induced by UVB irradiation and hydrogen peroxide in these cells. Specifically, the levels of ROS in HaCaT cells treated with OA-VI12 and subjected to H_2_O_2_ or UVB irradiation were assessed by analyzing four experimental groups: H_2_O_2_-treated cells, UVB-irradiated cells, a blank control group, and a sample group. The sample group comprised cells pretreated with varying concentrations of OA-VI12 at 37 °C for 2 h, followed by treatment with H_2_O_2_ for an additional 2 h or UVB irradiation. Subsequently, all groups were treated with the DCFH-DA fluorescent probe, and fluorescence intensity was measured using flow cytometry. The results indicated that HaCaT cells treated with the peptide and exposed to UVB and H_2_O_2_ exhibited a significant reduction in intracellular ROS fluorescence intensity after UVB irradiation and H_2_O_2_ stimulation. The efficacy of OA-VI12 on HaCaT cells was comparable to that of vitamin C. Utilizing commercial kits, we also quantified the levels of enzymes such as CAT and LDH in the treated sample groups, specifically in lysed HaCaT cells and the supernatants of the culture medium. This resulted in an increased release of CAT and reduced levels of LDH [11].

In addition, in vivo experiments were conducted to assess the photoprotective effect of OA-VI12 on the dorsal skin of mice subjected to UVB irradiation for 21 days. Observations were made regarding skin morphology, damage, erythema, and epidermal thickening. OA-VI12 effectively inhibited these phenomena, demonstrating beneficial activity compared to vitamin C. After UVB irradiation, the thickness of the epidermis significantly increased by 160.81 ± 6.93 μm in comparison to normal skin; however, treatment with OA-VI12 reduced this thickness to 62.14 ± 3.39 μm. Similarly, dermal thickness increased by 355.92 ± 8.28 μm following UVB irradiation, while OA-VI12 treatment decreased the thickness to 266.67 ± 6.36 μm. Additionally, histological analysis using Masson’s trichrome staining revealed that UVB irradiation diminished collagen fiber content, which was mitigated by treatment with OA-VI12 and vitamin C, reflecting similar anti-photodamage effects in both treatments. Tissue samples from the groups were homogenized in phosphate-buffered saline (PBS), and the supernatant was utilized to assess the levels of SOD and GSH using commercial kits. The results indicated that the peptide stimulated the production of SOD and GSH. These findings highlight the beneficial role of the AOPs encoded by the gene and the peptide’s potential application as a protective agent against photodamage [11].

In another study conducted on the skin secretions of *O. andersonii*, a peptide named OA-GL21 (GLLSGHYGRVVSTQSGHYGRG) was identified, which demonstrated weak antioxidant activity through ABTS^•+^ and DPPH free radical scavenging methods, although it exhibited high stability. However, this peptide significantly enhanced wound healing in human keratinocytes and fibroblasts in a dose- and time-dependent manner. In conclusion, this research demonstrated the effects of OA-GL21 on cellular damage and animal wounds and provided a novel template peptide for the development of wound-repair drugs [75].

### 2.15. Temporin Family

From a large group of AOP and AMP families present in three East Asian frog species (*Amolops lifanensis*, *Hylarana taipehensis*, and *Amolops granulosus*), the peptides temporin-TP1 (FLPVLGKVIKLVGGLL(NH_2_)), temporin-TP2 (FLPLLVGAISSILPKIF(NH_2_)), and temporin-TP3 (FLPLLFGALSTLLPKIF(NH_2_)) were found from *H. taipehensis;* temporin-LF1 (FLPFVGKLLSGLL(NH_2_)), and temporin-LF2 (FLPIVTGLLTSLL(NH_2_)) from *A. lifanensis;* temporin-1P (FLPIVGKLLSGLL(NH_2_)) and temporin-MT1 (FLPIVTGLLSSLL(NH_2_)) from the species *A. lifanensis* and *A. granulosus*; and temporin-CG3 (FLPIVGKLLSGLF(NH_2_)) only from the species *A. granulosus*. The ability of the temporin-TP1 peptide to scavenge ABTS^•+^ and/or DPPH radicals has been demonstrated [8,64]. Despite not exhibiting significant antimicrobial activity against certain microorganisms, temporin-TP1 showed strong antimicrobial effects (with a minimum inhibitory concentration, MIC = 3.1 mM). Although temporin-TP1 did not show apparent antimicrobial activity against some microorganisms, it exhibited significant antimicrobial activity (with MIC = 3.1 mM) against Gram-positive bacteria such as *Staphylococcus aureus* (ATCC 25923), *Enterococcus faecalis* 981 (IS), and *Nocardia asteroides* 201118 (IS) [32].

In a study that identified 50 peptides grouped into 21 peptide families exhibiting antioxidant and antimicrobial activity from *Amolops daiyunensis*, *Hylarana maosuoensis*, *Pelophylax hubeiensis*, and *Nanorana pleskei*, the AMPs temporin-DY1, temporin-HB1, temporin-HB2, temporin-MS1, and temporin-MS4 were found, varying significantly in their activities due to the great variety of peptide structures they present. Of the peptides found, temporin-MS1 (FLTGLIGGLMKALGK) was synthesized and showed a slight ABTS^•+^ free radical scavenging capacity, with 21.4 ± 2.2% eradication rates. In the antimicrobial activity results, temporin-MS1 showed activity against different Gram-positive strains, such as *Staphylococcus aureus* (ATCC 25923), *Enterococcus faecalis* 981 (IS), and *Nocardia asteroides* 201118 (IS), as well as Gram-negative strains, such as *Pseudomonas aeruginosa* (CGMCC 1.50), *Klebsiella pneumoniae* 08040724 (IS), and *Escherichia coli* (ATCC 25922). However, the peptides temporin-MS1 and temporin-MS4 exhibited the strongest hemolytic activity, which is an undesirable effect associated with these peptides [72].

### 2.16. Daiyunin and Pleskein Families

The daiyunin family is a group of peptides found in *Amolops daiyunensis*, an East Asian frog species from China. Here were identified daiyunin-1 (CGYKYGCMVKVDR), daiyunin-2 (FFGTKGIFSKVEPIFCKISHSC), and daiyunin-3 (IVRPPIRCKAAFC). Among these, daiyunin-1 exhibited significant results in the antioxidant and antimicrobial assays. At a concentration of 50 μM, the scavenging ability of daiyunin-1 against ABTS^•+^ and DPPH radicals within 30 min of the reaction was low, but its scavenging rate against ABTS^•+^ reached 77.1% when the reaction time was prolonged to 14 h. In the same study, from *Nanorana pleskei*, an East Asian frog species from China, pleskein-1 (FFPLIPGVRCKILRTC), pleskein-2 (FFLLPIPNDVKCKVLGICKS), and pleskein-3 (ILPSKLCRLLGNC) were reported. Antioxidant and antimicrobial activity assays showed that pleskein-1 and pleskein-2 have relatively weak antimicrobial activities, and only pleskein-2 has specific antioxidant activity (11.3% in an ABTS assay after 30 min at 50 μM). Pleskein-3 did not show any antimicrobial or antioxidant activity; however, it is inferred that pleskein-3, by sharing the same precursors with AMP or AOP, may possess other functions that help frogs adapt to their living environments [72].

### 2.17. Jindongenin Family

This family was reported in the Chinese torrent frog *Amolops jingdongensis*. One of its components is a 24-amino-acid peptide, jindongenin-1a (DSMGAVKLAKLLIDKMKCEVTKAC), tested for various activities, including antioxidant activity. Jindongenin-1a showed broad-spectrum antimicrobial activity against standard and clinically isolated bacterial strains. However, DPPH free radical scavenging assays indicated that jindongenin-1a exhibited low antioxidant activity at doses up to 32 mM (8% inhibition). The results suggest that these peptides do not play a key role in the habituation of *A. jingdongensis* to intense UV exposure [34].

### 2.18. Tryptophilins Family

As for tryptophilins, they are one of the first peptides identified in amphibians’ skin secretions. They are characterized as a heterogeneous group of peptides that were recently reclassified according to their structures into the following groups: T-1 (*C*-amidated heptapeptides and non-amidated octapeptides that have an *N*-terminus formed by Lys and Pro residues, a Trp residue at position 5, and a Pro residue at position 7), T-2 (four to seven amino acid residues, which have an internal Pro-Trp), and T-3 (tridecapeptides with five conserved Pro residues and the absence of Trp) [60]. Tryptophilins have been commonly associated with myoactive and vasorelaxation/vasoconstriction properties [60,62], opioid-like biological activities [61], antiproliferative effects [62], and antimicrobial actions [40,84].

Recent studies have found tryptophilin-like peptides with antioxidant potential in frogs of the species *Pithecopus azureus* [48] and *Pelophylax perezi* [76]. For *P. azureus*, the amidated AOP PaT-2 (FPPWL(NH_2_)) and its respective analogs PaT-2a1 (FPLPW(NH_2_)) and PaT-2a2 (PWLFP(NH_2_)) were reported [48]. For *P. perezi*, the also amidated antioxidant peptide PpT-2 (FPWLLS(NH_2_)) was reported, which demonstrated significant antioxidant potential for tryptophilin-like peptides [76]. The results of the analysis of peptides PaT-2 and its analogs (PaT-2a1 and PaT-2a2) showed antioxidant activity and low cytotoxicity in mammalian central nervous system cells such as mouse BV2 microglia and human neuroblastoma cells SK-N-BE(2), which were stimulated with the diester phorbol 12-myristate 13-acetate (PMA) and treated simultaneously with PaT-2, PaT-2a1, or PaT-2a2 at concentrations of 50 µM and 100 µM. ROS and RNS were detected before and after flow cytometry analysis for these cells, and they showed significant differences compared to control cells treated only in DMEM medium. Additionally, morphological and histological studies of the skin to identify the glands of *P. azureus*, as well as the results of the in silico and in vitro radical scavenging tests, allowed for verification of the possible relationship between PaT-2 and oxidative protection, leading the authors to conclude that they had identified, for the first time, a tryptophilin-like peptide with antioxidant potential [48].

On the other hand, the results from the analysis of the antioxidant peptide PpT-2 were measured based on the mechanism of action of these compounds, as it generally involves a series of complex processes for deactivating and trapping free radicals. These processes include the transfer of relevant charges and the antioxidant activities of biological systems. To evaluate the potential antioxidant properties of PpT-2, its ability to eliminate ABTS and DPPH radicals was assessed in vitro, using Trolox and GSH as reference compounds. The results indicated that PpT-2 exhibited an ABTS scavenging activity of 0.269 mg·Trolox-eq·mg^−1^ of peptide, comparable to other amphibian-derived peptides such as salamandrin-I (FAVWGCADYRGY(NH_2_)), which displayed a value of 0.285 mg·Trolox-eq·mg^−1^ of peptide, and antioxidin-RP1 (AMRLTYNKPCLYGT) showed 0.300 mg·Trolox-eq·mg^−1^ of peptide. In contrast, these values were significantly higher than that of antioxidin-I (TWYFITPYIPDK), with an activity of 0.010 mg·Trolox-eq·mg^−1^ of peptide [49], but much lower than GSH, which had a value of 1.911 mg·Trolox-eq·mg^−1^ of peptide. These results suggest that PpT-2 possesses free radical scavenging activities, demonstrating the strongest and most potent activity against ABTS radicals. PpT-2′s neuroprotective activity was evaluated using mammalian cells, specifically mouse (*Mus musculus*) microglia. These cells respond to tissue damage and infection by producing and releasing ROS and RNS into the surrounding central nervous system (CNS) environment. In this study, mouse microglial cells (BV-2) were stimulated with PMA, a chemical known to induce ROS and RNS production, either in the presence or absence of PpT-2 at two different concentrations. PMA activates protein kinase C, which phosphorylates p47phox to activate NADPH oxidase (NOX), leading to increased ROS generation. The results revealed that PpT-2 effectively inhibits oxidative stress in BV-2 cells at the tested concentrations during the 30 min of simultaneous incubation with PMA, right at the onset of treatment. Notably, PpT-2 also inhibits RNS generation even in cells not stimulated by PMA, indicating that this inhibition occurs under basal conditions. The results showed that PpT-2 regulates the steady-state levels of both ROS and RNS through its antioxidant effects. Thus, tryptophilin shows promise as a therapeutic agent for treating or preventing neurodegenerative disorders due to its antioxidant activity in microglia and the role of redox imbalance in the onset and progression of diseases such as Alzheimer’s, Parkinson’s, and Huntington’s. Despite the number of tryptophilins known, this is the first study to verify this bioactivity and therapeutic potential for PpT-2 [76].

### 2.19. Other AOPs from Amphibians

Some AOPs identified in amphibian skin are not associated with previously recognized large families of AMPs. Examples include peptides from unique families such as macrotympanin A1 (FLPGLECVW), hejiangin-A1 (RFIYMKGFGKPRFGKR), and wuchuanins (including wuchuanin-AOP5 (TVWGFRPSKPPSGYR) and wuchuanin-A1 (APDRPRKFCGILG)), which were found in the skin secretions of odorous frogs, including *Odorrana andersonii*, *O. rotadora*, *O. wuchuanensis*, and *O. margaritae*. These peptides demonstrated potent ABTS^•+^ radical scavenging activity, supporting the hypothesis that AOPs often contain free Cys residues crucial for their radical scavenging capabilities [59]. From the Xizang Plateau frog (*Nanorana parkeri*), parquerin (GWANTLKNVAGGLCKITGAA) was identified, exhibiting DPPH radical scavenging activity [33]. In another study, the antioxidant peptides ranacyclin-HB1 (GAPKGCWTKSYPPQPCFGKK) and odorranaopin-MS2 (DNVYSRPPQRFGQNVIS), identified from the skin secretions of *Amolops daiyunensis*, *Pelophylax hubeiensis*, *Nanorana pleskei*, and *Hylarana maosuoensis*, demonstrated strong ABTS and DPPH free radical scavenging abilities [72].

From skin secretion samples of the salamander (*Salamandra salamandra*), the peptide salamandrin-I (FAVWGCADYRGY(NH_2_)) was identified as a potential antioxidant, capable of scavenging DPPH and ABTS^•+^ free radicals [79]. Another peptide, ansin-2 (TRCFRVCS), was designed de novo from natural short AOPs derived from frogs and demonstrated a protective effect against UV-induced sunburn and hyperpigmentation in guinea pig skin models following topical administration [85].

Another type of AOP consists of those resulting from the enzymatic hydrolysis of bullfrog muscle protein (*Rana catesbeiana* Shaw). An example is APBMH (LEQQVDDLEGSLEQEKK). After purification, its free radical scavenging activity was confirmed through electron spin resonance (ESR) spectroscopy and a polyunsaturated fatty acid (PUFA) peroxidation inhibition assay. APBMH demonstrated scavenging activity comparable to that of vitamin C against DPPH, hydroxyl, and superoxide radicals [78]. Another AOP identified through similar methods was APBSP (LEELEEELEGCE), which exhibited a more significant inhibition of lipid peroxidation than α-tocopherol, serving as a positive control, and efficiently quenched various free radicals: DPPH, hydroxyl, superoxide, and peroxyl radicals [86]. Finally, peptides purified from frog protein hydrolysates derived from *Hylarana guentheri* (FPH) included Leu/Ile-Lys and Phe-Lys, both of which demonstrated significant activity in scavenging DPPH radicals, as well as in FRAP and ORAC assays [81].

## 3. Amphibian AOPs and Oxidative Stress-Related Diseases

Research has shown that an amino acid type and sequence influence a peptide’s antioxidant activity. Amphibian-derived AOPs can scavenge oxygen free radicals, activate the antioxidant enzyme system, inhibit DNA damage and apoptosis, and regulate the mitogen-activated protein kinase (MAPK) signaling pathway. However, their precise functions remain unclear [4]. Studies of model peptides suggest that any peptide satisfying the appropriate hydrophobicity and cationic criteria and capable of adopting amphipathic-helical conformation will display at least some antimicrobial [87], and antioxidant activity [8]. Furthermore, research on AOPs from Asian amphibians has shown that these peptides have precursors similar to AMPs [8,64], and many mature AOPs also exhibit antimicrobial activities, demonstrating a close evolutionary relationship [32], and maybe a similar mechanism of action. Additionally, it is well established that oxidative stress (OS) significantly impacts cell homeostasis and that excess ROS are linked to various diseases. While the pathophysiology of conditions such as cancer, neurodegenerative disorders (NDs), and cardiovascular diseases differs, OS remains a common factor among them [88]. By scavenging free radicals, AOPs may prevent or mitigate OS and help restore redox balance, which is crucial for treating these complex diseases. OS is known to cause damage to cellular components, inflammation, mitochondrial dysfunction, and activation of cell death pathways in many NDs, or even resistance to cell death in cancers (Figure 1) [89,90,91,92]. Therefore, given the importance of antioxidant treatments in this context, this section will reference reports of amphibian AOPs with antioxidant activities and other mechanisms of action evaluated directly against OS-related diseases.

The brain’s physiology makes it considerably more susceptible to OS damage. Its high energy demand, high oxygen consumption, high polyunsaturated fatty acid content, and excessive iron concentration are reasons for that susceptibility [93]. This increased sensitivity to OS makes antioxidants an essential adjunct in ND treatment, and there are reports of the potential of amphibian AOPs with neuroprotective effects. Brevinin-1FL, an AOP from the skin of *Fejervarya limnocharis*, exhibited excellent antioxidant activity in vitro through its high reducing power and radical scavenging activities, as well as anti-inflammatory potential by reducing the secretion of pro-inflammatory cytokines and increasing levels of antioxidant enzymes. Brevinin-1FL also presents neuroprotective effects, as it reverses the H_2_O_2_-induced changes in PC-12 cells [71]. PaT-2, a peptide from *Pithecopus azureus*, prevented oxidative stress in the human microglia cell model by hampering lipopolysaccharide-induced ROS and glutamate release, which is known to influence neuronal cell death [48]. These results support amphibian AOPs as a potential source of molecules to aid in ND treatment: microglial cells are important targets for antioxidant treatments because OS and inflammation related to these cells play essential roles in NDs such as Alzheimer’s disease (AD), Parkinson’s disease (PD), amyotrophic lateral sclerosis (ALS), and others [94,95].

AD is the most common ND and presents several hallmarks targeted for drug discovery, such as the formation of amyloid-beta (Aβ) plaques, tau protein depositions, and altered acetylcholinesterase (AChE) activity. The use of AChE inhibitors, including antioxidants and multi-target compounds, is one of the primary strategies for combating AD, as most drugs prescribed for AD treatment act by inhibiting AChE [96]. Additionally, there are reports of AOPs exhibiting in vitro activities against pathways related to AD. A study on peptide skin extracts from three different amphibian families in the Argentinean Litoral region found that five out of nine evaluated extracts had the potential to act on four critical pathways of AD: inhibition of AChE and butyrylcholinesterase (BChE), which are responsible for low levels of acetylcholine in AD, and monoamine oxidase B (MAO-B), known to improve AD symptoms, as well as exhibiting antioxidant potential due to free radical scavenging activity [97]. More recently, this same group also reported that the natural peptide Bcl-1003 (GSKKTKCPR(NH_2_)), isolated from the skin of *Boana cordobae*, inhibits both BChE and MAO-B and exhibits antioxidant activity [98]. Among AChE inhibitors, those that target the peripheral anionic site (PAS) are of great interest. In one study, a 10-amino acid peptide called LL (LGPVSKGKLL(NH_2_)) was derived from a more prominent natural sequence called Hp-1935, isolated from the skin secretions of the Argentine frog *Boana pulchella*, which exhibits PAS inhibitory activity [99]. Since peptide LL is a PAS inhibitor, it was complexed with AChE through a flexible docking study, and the results were corroborated with 100 ns of molecular-dynamic simulations. The LL-AChE complex involves three critical residues of the PAS (Tyr-70, Trp-279, and Tyr-334), all interacting through hydrophobic interactions with Pro-3 in peptide LL. Because Pro-3 was influential in the docking analysis, a rational design and synthesis of LL peptide analogs based on the substitution of Pro-3 alone, or Pro-3 and aliphatic residues, such as Gly-7 in the *C*-terminus, with aromatic amino acids (Trp, Tyr, Phe, and 4-fluoro-Phe) was proposed to form π–π stacking interactions with the aromatic residues of the PAS. Finally, the authors compared the inhibitory and antioxidant properties of the new sequences with LL to evaluate the effect of the modifications. Among the analogs, peptide W3 (LGWVSKGKLL(NH_2_)) has an IC_50_ of 10.42 μM and was found to exhibit a 30-fold enhancement in AChE inhibitory activity, along with antioxidant activity, making it one of the most potent against this Alzheimer’s-related enzyme. Additionally, it was observed that W3 interacts with AChE in the same region as peptide LL, specifically the PAS. Molecular mechanics analysis indicated that Trp-3 of W3 is located close to the three critical residues of the PAS site throughout the simulation time, where the indole side chain of W3 is sandwiched between residues Tyr-121, Trp-279, and Tyr-334 of the PAS through aromatic–aromatic stacking interactions (Figure 2) [80].

Parkinson’s disease (PD) is caused by the degeneration of dopaminergic neurons in the midbrain region, also known as the substantia nigra, characterized by the accumulation of α-synuclein [100,101,102]. Studies show that excessive ROS leads to the oxidation of macromolecules, including lipids, proteins, and nucleic acids, and their accumulation in the brain tissues of Parkinson’s patients, causing local or systemic damage and promoting dopaminergic neural degeneration [102,103]. The previously mentioned AOP, PpT-2, has electron donation and acceptance capacity similar to other antioxidants, eliminates free radicals, and inhibits PMA-induced OS. It has also demonstrated neuroprotective potential in vitro on mouse microglial cells (BV-2) by controlling the levels and steady state of ROS and RNS. Therefore, PpT-2 could be a promising therapeutic agent for AD and PD [76]. Another study described Cath-KP, a peptide that belongs to the cathelicidin family, extracted from the skin of the Asian-painted frog *Kaloula pulchra*. Cath-KP penetrates cells and reaches deep brain tissues, resulting in improved cell viability even when treated with MPP^+^, and it reduces OS-induced damage by inducing the expression of antioxidant enzymes and alleviating the accumulation of mitochondrial and intracellular ROS. In mice with PD induced by MPTP, there was an increase in tyrosine hydroxylase (TH)-positive neurons, improving dyskinesia. Cath-KP also alleviated the accumulation of mitochondrial and intracellular ROS by activating the sirtuin-1 (Sirt1)/nuclear factor erythroid 2-related factor 2 (Nrf2) pathway, with focal adhesion kinase (FAK) and p38 identified as regulatory elements. Thus, this demonstrates that Cath-KP can prevent damage caused by OS due to its antioxidant and neuroprotective properties [69].

Mitochondria play a crucial role in apoptosis, a programmed cell death process, and MAPK signaling pathways can act as either activators or inhibitors of apoptosis. Tests with antioxidant peptides in cell lines have allowed us to identify some mechanisms of action that demonstrate their effect on signaling pathways involved in cell proliferation and translocation to mitochondria. For example, peptide OA-VI12, identified in the frog *O. andersonii* [11], can regulate the MAPK signaling pathway [104], which is usually activated by oxidative stress and is related to cell proliferation and migration, signal transduction, inflammation, collagen synthesis, and apoptosis. Additionally, in the research on apoptotic mechanisms of AOPs, it is commonly expected to detect levels of crucial apoptotic proteins, gene expression of Bcl-2 family proteins, and the translocation of cytochrome C (Cyt C) into mitochondria and the cytoplasm [105]. OM-GL15 (GLLSGHYGRASPVAC) from the skin of the green odorous frog, *O. margaretae*, is an AOP that protects epidermal cells from apoptosis, significantly attenuating UVB radiation-induced skin photodamage in mice by reducing lipid peroxidation and MDA levels through upregulation of Bcl-2; inhibition of DNA damage; and downregulation of caspase-3, caspase-9, and Bax [31]. Antioxidin-RL can regulate the accumulation of free radicals and the expression of superoxide dismutase-1 (SOD1) and glutathione peroxidase-1 (GPx1) in rat PC-12 cells, where oxidation is induced with H_2_O_2_, while maintaining mitochondrial morphology to ensure cellular metabolism [72]. Similarly, brevinin-1FL was able to reverse H_2_O_2_-induced oxidation in a cell-dependent manner in PC-12 cells, suggesting a protective effect against H_2_O_2_-induced damage to the mitochondrial membrane [71]. As mentioned above, peptides containing Pro and Cys are associated with the ability to eliminate oxygen free radicals, as is the case with this peptide that regulates the redox environment in mammalian cells [82]. OA-VI12, OM-GL15, and antioxidin-RL contain these amino acids.

Several reports of amphibian peptides describe activity against numerous cancers, with most exhibiting antitumor activity by killing cells, affecting proliferation, modulating the immune system, or inhibiting angiogenesis [106]. There is a well-established and widely described relationship between OS and cancer in the literature [92,107], demonstrating that OS can lead to DNA damage and consequent mutations, overactivation of signaling pathways related to cell growth and proliferation, modulation of the tumor microenvironment, and promotion of cell survival and resistance to therapies [108,109,110,111]. Therefore, in theory, controlling OS seems to be an essential step not only for cancer treatment but also for prevention since chronic OS can favor tumor growth and progression [112]. However, there are still some limitations regarding using antioxidants for cancer treatment. Some key issues may influence the low efficacy of antioxidants in clinical practice. First, reliance on pharmacological doses in studies rather than dietary doses does not accurately reflect in vivo conditions. Additionally, antioxidants may not be evenly distributed throughout the body. They can have low bioavailability in specific tissues, and their effects can also vary depending on concentration and oxygen levels, shifting between antioxidant and pro-oxidant properties. Finally, since many chemotherapy drugs generate ROS, antioxidants could potentially interfere with the effectiveness of these treatments, undermining their ability to induce cancer cell death [113].

Although there are many controversies, acknowledging these limitations might help optimize the design of assays and studies in which adding antioxidants could be advantageous. Additionally, drug delivery strategies, such as nanoparticles, are alternatives that could help overcome some of these issues [114,115]. The development of new cancer therapies is necessary not only to help overcome resistance problems but also to improve the quality of life for cancer patients, relieving symptoms and perhaps even preventing recurrence. Different antioxidant molecules might present distinct results, so seeking, characterizing, and evaluating new antioxidant molecules and sources, such as amphibian AOPs, may constitute a possibility for developing adjunctive cancer treatments; however, further investigation is necessary.

The skin is regularly exposed to OS from endogenous and exogenous sources. Air pollution, chemicals, microorganisms, toxins, harmful natural gases, and ionizing and non-ionizing radiation are examples of exogenous agents that contribute to ROS generation and consequent OS in the skin [114,115]. Maintaining skin homeostasis is essential and particularly challenging, mainly due to its significant exposure to UV radiation, which does not affect other organs [115]. UV radiation impacts both the skin’s structure and functionality, and excessive exposure can result in immediate and long-lasting harmful effects, including photodamage, premature aging, and an increased risk of skin cancer [116,117].

OS-LL11 (LLPPWLCPRNK) was identified from *O. schmackeri*. This AOP can scavenge free radicals and exhibit antioxidant, anti-inflammatory, and anti-apoptotic properties. In vitro, results demonstrated that the peptide was able to preserve the cell viability of mouse keratinocytes treated with UVB irradiation or H_2_O_2_ by diminishing harmful effects, such as lipid peroxidation and ROS, while increasing the levels of CAT and other essential proteins involved in cellular responses to oxidative stress and regulation of antioxidant defenses. OS-LL11 activity was also evaluated in vivo, and the photoprotective effects on mouse skin occurred due to upregulation of superoxide dismutase, glutathione, and nitric oxide levels, as well as a reduction in the number of apoptotic bodies and downregulation of H_2_O_2_, IL-1α, IL-1β, IL-6, TNF-α, and other proteins involved in OS and apoptosis [77].

The activity of the AOP, OA-GI13 (GIWAPWPPRAGLC), identified from *O. andersonii*, was described as exhibiting free radical scavenging activity and was able to maintain keratinocyte viability under treatment with H_2_O_2_ by protecting cell membrane integrity, as evidenced by reduced LDH levels in the presence of the peptide, and also due to the stimulation of SOD, CAT, and GSH release [117]. The activity of OA-GI13 was also tested in mice. The peptide exhibited photoprotective action against UVB radiation exposure, as evidenced by decreased erythema and edema, and reduced levels of OS markers, like peroxide. Another mechanism by which the peptide exerts antioxidant activity was observed in vitro and in vivo. The inhibition of p38 phosphorylation upon OA-GI13 treatment suppresses the p38 MAPK signaling pathway and prevents ROS production. Other studies describe amphibian AOPs with photoprotective activity, which supports their applicability since they recover redox balance and even prevent UV radiation’s detrimental effects by maintaining homeostasis. Their potential against acute symptoms is established, and they constitute a basis for further investigation regarding protection against chronic effects of UV exposure, such as skin cancer and aging [11,42,118].

Skin regeneration is another process of interest in drug development, as it is crucial for maintaining homeostasis and is also affected by diseases. Amphibians, due to their high wound-healing capacity [119] and their ability to regenerate limbs or tails, especially in young animals, are promising sources of molecules with regenerative properties. There are many reports of amphibian-derived wound-healing peptides [120], but only one, to our knowledge, with both antioxidant and regenerative activity. Cathelicidin-OA1, previously mentioned in this review, was the first amphibian peptide of the cathelicidin group to show solid wound-healing activity. In vitro, it improved wound healing in human keratinocytes and fibroblasts. In vivo, the peptide increased macrophage recruitment to the wound site, stimulating cell proliferation, migration and increasing re-epithelialization and granulation tissue formation [67]. Since OS interferes with wound repair, cathelicidin-OA1 promotes regeneration, directly facilitating the healing process and enhancing repair capacity through its antioxidant activity.

As previously discussed, OS is a common factor among various diseases. In this review, we focused on conditions for which amphibian AOPs or amphibian peptides have already been reported; however, their comparison against other OS-related diseases is also valid. Other neurodegenerative disorders, such as Huntington’s disease (HD) and amyotrophic lateral sclerosis (ALS), are greatly affected by OS, as are cardiovascular and metabolic diseases. There are reports of peptides with antioxidant activity from different sources with multifunctional properties, i.e., AOPs displaying multiple activities that can be of interest for treating multifactorial diseases [121,122].

## 4. Future Prospects

Amphibian AMPs, mainly AOPs, due to their antioxidant activity, show significant promise for addressing diseases related to oxidative stress, such as cancer, neurodegenerative disorders, cardiovascular disorders, and other diseases. These peptides can act as direct scavengers of ROS and RNS species, and this potential, together with their antimicrobial properties, positions them as valuable multifunctional molecules. Although some of these peptides present amino acids in common that are attributed to the ability to stabilize free radicals, such as Trp, Tyr, Pro, and Phe, the structural diversity of these peptides poses challenges, as the lack of identical sequences in all families emphasizes the need for extensive research to decode their unique structures and functions.

The studies highlight the complex mechanisms through which amphibian peptides mitigate oxidative stress. They regulate levels of reactive species associated with oxidative stress through unclear enzymatic and non-enzymatic pathways. Elucidating these mechanisms is particularly relevant because it enhances our understanding of how these diseases function, thereby enabling the design of analogous peptides to treat conditions for which oxidative stress is a common underlying factor.

Investigating the interactions of amphibian peptides—not solely through computational modeling at the atomic level but also by exploring the intricate mechanisms by which these molecules function within biological environments linked to diseases—enhances our understanding of their roles. This insight paves the way for developing effective, nature-inspired therapeutic agents. Future research directions should focus on comprehensive characterization involving systematic studies to characterize the structures and functions of various amphibian AOPs and AMPs in different species; mechanistic studies with in-depth investigations into the molecular mechanisms of action of these peptides, especially in vivo models, to validate their therapeutic potential; and translational research exploring the clinical applications of these peptides, including their integration into current treatment regimens for oxidative stress-related diseases. Through a multidisciplinary approach leveraging advances in biochemistry, pharmacology, and molecular biology, the potential of amphibian-derived peptides could be harnessed to create innovative therapeutic solutions that address some of the most pressing health challenges related to oxidative stress.

## 5. Conclusions

Amphibians have evolved a complex array of peptides crucial for survival, particularly in response to environmental stressors. AOPs play a vital role in alleviating oxidative stress through free radical scavenging, metal chelation, and activating endogenous antioxidant systems. Their significance extends beyond antioxidative functions, as many AOPs also possess antimicrobial properties, highlighting their multifunctional capabilities. Despite the benefits presented by these peptides, significant knowledge gaps remain regarding their structural and functional relationships and their precise roles in various biological processes. Further studies are essential to unravel the molecular mechanisms of action associated with AOPs, enhancing our understanding of their therapeutic potential. Continued research could lead to the development of effective antioxidant therapies.

The ongoing exploration of amphibian-derived peptides holds promise for advancing innovative treatments for numerous health issues related to oxidative stress. Recognizing the multifaceted functions of these peptides will be critical in harnessing their potential for therapeutic applications in clinical settings.

## Figures and Tables

**Figure 1 antibiotics-14-00126-f001:**
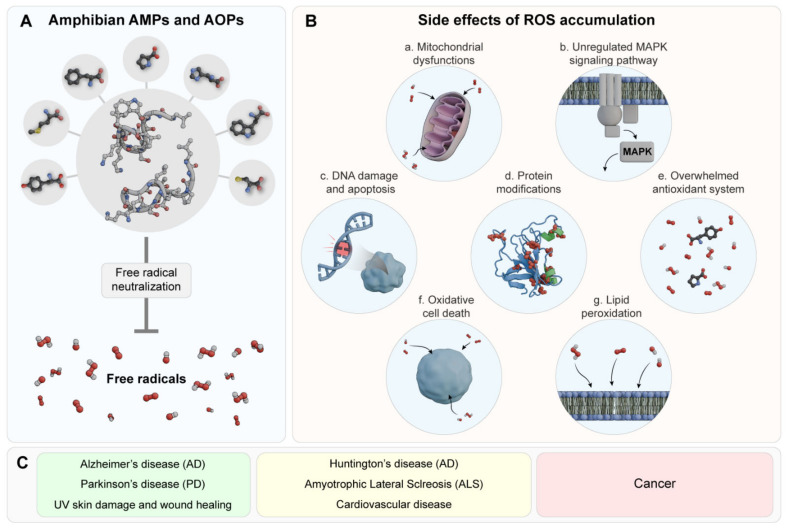
Overview of how AMPs from amphibians can act to neutralize oxidative stress, mitigate the side effects of ROS, and illustrate the relationship between excess ROS and certain neurodegenerative diseases and age-related conditions (such as cardiovascular disease and cancer), for which AOPs can act as mediators. (**A**) AOPs are predominantly free radical scavenging AMPs, which are inhibitors of free radicals. This property is attributed to the nature of specific amino acids, such as Phe, Tyr, Cys, Pro, Trp, and His, which, when present within sequences, can confer antioxidant properties. However, other AMPs may exhibit different mechanisms of action and have varying effects on these diseases. (**B**) Side effects of oxidative stress (OS), where excess ROS can lead to harmful consequences at the level of organelles, biomolecules, and signaling pathways, including (**a**) mitochondrial dysfunctions, (**b**) unregulated MAPK signaling pathways, (**c**) DNA damage and apoptosis, (**d**) protein modifications, (**e**) overwhelmed antioxidant systems, (**f**) oxidative cell death, (**g**) lipid peroxidation, and other unmentioned alterations causing an imbalance in cellular homeostasis. (**C**) These disturbances and alterations are directly involved in the development of human diseases, including Alzheimer’s disease (AD), Parkinson’s disease (PD), Huntington’s disease (HD), amyotrophic lateral sclerosis (ALS), UV skin damage and wound healing, cardiovascular diseases, and cancer, all related to an excess of ROS. Preventing such consequences is important for treating or preventing oxidative stress-related diseases. AD, PD, and UV skin damage and wound healing are conditions for which some amphibian AOPs have already been tested. HD, ALS, and cardiovascular diseases are some of the medically important OS-related diseases for which amphibian AOPs could potentially act. Cancer is a disease for which antioxidants could theoretically serve as adjuvant treatments. Although the AOPs described so far do not show significant importance as therapy, evaluating the potential of new molecules could lead to different outcomes is crucial.

**Figure 2 antibiotics-14-00126-f002:**
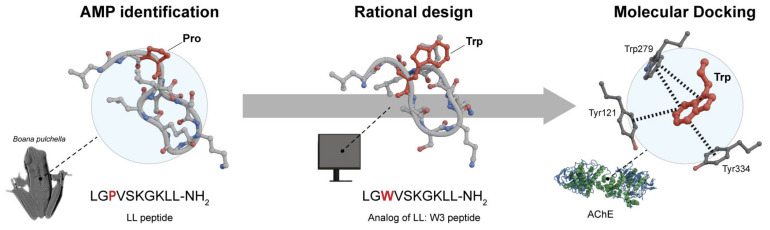
Molecular inhibition mechanism of peptide W3 against acetylcholinesterase (AChE). Peptide W3 is a potent inhibitor of AChE and an analogue of peptide LL, identified from the frog *Boana pulchella* (amphibian). Molecular docking shows that the Trp-3 residue at the *N*-terminus of peptide W3 generates π–π stacking interactions with the aromatic residues (Tyr-121, Trp-279, and Tyr-334) of the peripheral anionic site (PAS) of AChE.

**Table 1 antibiotics-14-00126-t001:** Amphibian AOPs are grouped by families, showing the sequences, methods for evaluating antioxidant activity, and some amino acids responsible for this activity (Pro, Tyr, Trp, Met, and Cys).

AOPs Families	Name	Sequence as 1- and 3-Letter Symbols	Number of Amino Acids	Methods Used for Antioxidant Activity	Possible Functional Amino Acids	References
Antioxidin	Antioxidin-RP1	AMRLTYNKPCLYGT Ala-Met-Arg-Leu-Thr-Tyr-Asn-Lys-Pro-Cys-Leu-Tyr-Gly-Thr	14	Free radical scavenging activity: DPPH and ABTS^•+^; FRAP.	1 Pro 2 Tyr 1 Met 1 Cys	[8]
	Antioxidin-RP2	SMRLTYNKPCLYGT Ser-Met-Arg-Leu-Thr-Tyr-Asn-Lys-Pro-Cys-Leu-Tyr-Gly-Thr	14	Free radical scavenging activity: DPPH and ABTS^•+^; FRAP.	1 Pro 2 Tyr 1 Met 1 Cys	[8]
	Antioxidin-RL	AMRLTYNRPCIYAT Ala-Met-Arg-Leu-Thr-Tyr-Asn-Arg-Pro-Cys-Ile-Tyr-Ala-Thr	14	Free radical scavenging activity: ABTS^•+^ and FRAP; ABTS^•+^ reaction kinetics.	1 Pro 2 Tyr 1 Met 1 Cys	[64]
	Antioxidin-PN	FLPSSPWNEGTYVLKKLKS Phe-Leu-Pro-Ser-Ser-Pro-Trp-Asn-Glu-Gly-Thr-Tyr-Val-Leu-Lys-Lys-Leu-Lys-Ser	19	Free radical scavenging activity: ABTS^•+^ and ABTS^•+^ reaction kinetics.	2 Pro 1 Tyr 1 Trp	[65]
	Antioxidin-2	YMRLTYNRPCIYAT Tyr-Met-Arg-Leu-Thr-Tyr-Asn-Arg-Pro-Cys-Ile-Tyr-Ala-Thr	14	Free radical scavenging activity: ABTS^•+^; attenuated PC-12 cell injury induced by H_2_O_2_; suppress intracellular ROS content boosted by H_2_O_2_; inhibition of the change in SOD1 and GPx1 expression caused by H_2_O_2_ stimulation.	1 Pro 3 Tyr 1 Met 1 Cys	[66]
	Antioxidin-I	TWYFITPYIPDK Thr-Trp-Tyr-Phe-Ile-Thr-Pro-Tyr-Ile-Pro-Asp-Lys	12	In vitro radical scavenging assays: ABTS^•+^, DPPH, ORAC, and NO; GSH redox balance experiment.	2 Pro 2 Tyr 1 Trp	[49]
	Antioxidin-NV	GWANTLKNVAGGLCKMTGAA Gly-Trp-Ala-Asn-Thr-Leu-Lys-Asn-Val-Ala-Gly-Gly-Leu-Cys-Lys-Met-Thr-Gly-Ala-Ala	20	Free radical scavenging activity: ABTS^•+^; intracellular and mitochondrial ROS production (2′,7′-dichlorodihydrofluorescein diacetate (DCFH-DA) assay and kit assay, respectively); hairless mouse model of photoaged skin-immunohistochemistry analysis.	1 Trp 1 Met 1 Cys	[35]
Pleurain	Pleurain-A1	SIITMTKEAKLPQLWKQIACRLYNTC Ser-Ile-Ile-Thr-Met-Thr-Lys-Glu-Ala-Lys-Leu-Pro-Gln-Leu-Trp-Lys-Gln-IleAla-Cys-Arg-Leu-Tyr-Asn-Thr-Cys	26	Free radical scavenging activity: DPPH, ABTS^•+^, NO, and FRAP.	1 Pro 1 Tyr 1 Trp 1 Met 2 Cys	[8]
	Pleurain-D1	FLSGILKLAFKIPSVLCAVLKNC Phe-Lys-Ser-Gly-Ile-Leu-Lys-Leu-Ala-Phe-Lys-Ile-Pro-Ser-Val-Leu-Cys-Ala-Val-Leu-Lys-Asn-Cys	23	Free radical scavenging activity: DPPH, ABTS^•+^, NO, and FRAP.	1 Pro 2 Cys	[8]
	Pleurain-E1	AKAWGIPPHVIPQIVPVRIRPLCGNV Ala-Lys-Ala-Trp-Gly-Ile-Pro-Pro-His-Val-Ile-Pro-Gln-Ile-Val-Pro-Val-Arg-Ile-Arg-Pro-Leu-Cys-Gly-Asn-Val	26	Free radical scavenging activity: DPPH, ABTS^•+^, NO, and FRAP.	5 Pro 1 Trp 1 Cys	[8]
	Pleurain-G1	GFWDSVKEGLKNAAVTILNKIKCKISECPPA Gly-Phe-Trp-Asp-Ser-Val-Lys-Glu-Gly-Leu-Lys-Asn-Ala-Ala-Val-Thr-Ile-Leu-Asn-Lys-Ile-Lys-Cys-Lys-Ile-Ser-Glu-Cys-Pro-Pro-Ala	31	Free radical scavenging activity: DPPH, ABTS^•+^, NO, and FRAP.	2 Pro 1 Trp 2 Cys	[8]
	Pleurain-J1	FIPGLRRLFATVVPTVVCAINKLPPG Phe-Ile-Pro-Gly-Leu-Arg-Arg-Leu-Phe-Ala-Thr-Val-Val-Pro-Thr-Val-Val-Cys-Ala-Ile-Asn-Lys-Leu-Pro-Pro-Gly	26	Free radical scavenging activity: DPPH, ABTS^•+^, NO, and FRAP.	4 Pro 1 Cys	[8]
	Pleurain-K1	DDPDKGMLKWKNDFFQEF Asp-Asp-Pro-Asp-Lys-Gly-Met-Leu-Lys-Trp-Lys-Asn-Asp-Phe-Phe-Gln-Glu-Phe	18	Free radical scavenging activity: DPPH, ABTS^•+^, NO, and FRAP.	1 Pro 1 Trp 1 Met	[8]
	Pleurain-M1	GLLDSVKEGLKKVAGQLLDTLKCKISGCTPA Gly-Leu-Leu-Asp-Ser-Val-Lys-Glu-Gly-Leu-Lys-Lys-Val-Ala-Gly-Gln-Leu-Leu-Asp-Thr-Leu-Lys-Cys-Lys-Ile-Ser-Gly-Cys-Thr-Pro-Ala	31	Free radical scavenging activity: DPPH, ABTS^•+^, NO, and FRAP.	1 Pro 2 Cys	[8]
	Pleurain-N1	GFFDRIKALTKNVTLELLNTITCKLPVTPP Gly-Phe-Phe-Asp-Arg-Ile-Lys-Ala-Leu-Thr-Lys-Asn-Val-Thr-Leu-Glu-Leu-Leu-Asn-Thr-Ile-Thr-Cys-Lys-Leu-Pro-Val-Thr-Pro-Pro	30	Free radical scavenging activity: DPPH, ABTS^•+^, NO, and FRAP.	3 Pro 1 Cys	[8]
	Pleurain-P1	SFGAKNAVKNGLQKLRNQCQANNYQGPFCDIFKKNP Ser-Phe-Gly-Ala-Lys-Asn-Ala-Val-Lys-Asn-Gly-Leu-Glu-Lys-Leu-Arg-Asn-Gln-Cys-Gln-Ala-Asn-Asn-Tyr-Gln-Gly-Pro-Phe-Cys-Asp-Ile-Phe-Lys-Lys-Asn-Pro	36	Free radical scavenging activity: DPPH, ABTS^•+^, NO, and FRAP.	2 Pro 2 Cys 1 Tyr	[8]
	Pleurain-R1	CVHWMTNTARTACIAP Cys-Val-His-Trp-Met-Thr-Asn-Thr-Ala-Arg-Thr-Ala-Cys-Ile-Ala-Pro	16	Free radical scavenging activity: DPPH, ABTS^•+^, NO, and FRAP.	1 Pro 1 Trp 2 Cys 1 Met	[8]
	Pleurain-E-OT	ATAWRMPPNGIPPIVAVRIRPLCGTV Ala-Thr-Ala-Trp-Arg-Met-Pro-Pro-Asn-Gly-Ile-Pro-Pro-Ile-Val-Ala-Val-Arg-Ile-Arg-Pro-Leu-Cys-Gly-Thr-Val	26	Free radical scavenging activity: DPPH.	5 Pro 1 Cys 1 Trp 1 Met	[58]
Catelicidin	Catelicidin-OA1	IGRDPTWSHLAASCLKCIFDDLPKTHN Ile-Gly-Arg-Asp-Pro-Thr-Trp-Ser-His-Leu-Ala-Ala-Ser-Cys-Leu-Lys-Cys-Ile-Phe-Asp-Asp-Leu-Pro-Lys-Thr-His-Asn	27	Free radical scavenging activity: DPPH; ABTS^•+^.	2 Pro 1 Trp 2 Cys	[67]
	Catelicidin-NV	ARGKKECKDDRCRLLMKRGSFSYV Ala-Arg-Gly-Lys-Lys-Glu-Cys-Lys-Asp-Asp-Arg-Cys-Arg-Leu-Leu-Met-Lys-Arg-Gly-Ser-Phe-Ser-Tyr-Val	24	Free radical scavenging activity: ABTS^•+^; determination of intracellular ROS generation. Antioxidant enzyme assays in vivo (CAT and SOD); MDA level determination.	1 Tyr 1 Cys 1 Met	[68]
	Cath-KP	GCSGRFCNLFNNRRPGRLTLIHRPGGDKRTSTGLIYV Gly-Cys-Ser-Gly-Arg-Phe-Cys-Asn-Leu-Phe-Asn-Asn-Arg-Arg-Pro-Gly-Arg-Leu-Thr-Leu-Ile-His-Arg-Pro-Gly-Gly-Asp-Lys-Arg-Thr-Ser-Thr-Gly-Leu-Ile-Tyr-Val	37	Free radical scavenging activity: DPPH, ABTS^•+^, and FRAP; intracellular ROS detection of lives cells (DCFH-DA assay) by flow cytometry; intracellular SOD, CAT, and NO detection assays.	2 Pro 2 Cys 1 Tyr	[69]
Spinosan	Spinosan A	DLGKASYPIAYS Asp-Leu-Gly-Lys-Ala-Ser-Tyr-Pro-Ile-Ala-Tyr-Ser	12	Free radical scavenging activity: DPPH.	1 Pro 2 Tyr	[70]
	Spinosan B	DYCKPEECDYYFSFPI Asp-Tyr-Cys-Lys-Pro-Glu-Glu-Cys-Asp-Tyr-Tyr-Phe-Ser-Phe-Pro-Ile	16	Free radical scavenging activity: DPPH.	2 Pro 2 Cys 3 Tyr	[70]
	Spinosan C	DLSMMRKAGSNIVCGLNGLC Asp-Leu-Ser-Met-Met-Arg-Lys-Ala-Gly-Ser-Asn-Ile-Val-Cys-Gly-Leu-Asn-Gly-Leu-Cys	20	Free radical scavenging activity: DPPH.	2 Met 2 Cys	[70]
	Spinosan D	MEELYKEIDDCVNYGNCKTLKLM Met-Glu-Glu-Leu-Tyr-Lys-Glu-Ile-Asp-Asp-Cys-Val-Asn-Tyr-Gly-Asn-Cys-Lys-Thr-Leu-Lys-Leu-Met	23	Free radical scavenging activity: DPPH.	2 Met 2 Tyr 2 Cys	[70]
Palustrin	Palustrin-2AJ1	GFMDTAKNVAKNVAVTLIDKLRCKVTGGC Gly-Phe-Met-Asp-Thr-Ala-Lys-Asn-Val-Ala-Lys-Asn-Val-Ala-Val-Thr-Leu-Ile-Asp-Lys-Leu-Arg-Cys-Lys-Val-Thr-Gly-Gly-Cys	29	Free radical scavenging activity: DPPH.	2 Cys 1 Met	[34]
	Palustrin-2GN1	GLWNTIKEAGKKFALNLLDKIRCGIAGGCKG Gly-Leu-Trp-Asn-Thr-Ile-Lys-Glu-Ala-Gly-Lys-Lys-Phe-Ala-Leu-Asn-Leu-Leu-Asp-Lys-Ile-Arg-Cys-Gly-Ile-Ala-Gly-Gly-Cys-Lys-Gly	31	Free radical scavenging activity: DPPH and ABTS^•+^; ABTS^•+^ and DPPH free radical scavenging kinetics.	1 Trp 1 Cys	[32]
Brevinin	Brevinin-1TP1	FLPGLIKAAVGVGSTILCKITKKC Phe-Leu-Pro-Gly-Leu-Ile-Lys-Ala-Ala-Val-Gly-Val-Gly-Ser-Thr-Ile-Leu-Cys-Lys-Ile-Thr-Lys-Lys-Cys	24	Free radical scavenging activity: DPPH and ABTS^•+^; ABTS^•+^ and DPPH free radical scavenging kinetics.	1 Pro 2 Cys	[32]
	Brevinin-1TP2	FLPGLIKAAVGIGSTIFCKISKKC Phe-Leu-Pro-Gly-Leu-Ile-Lys-Ala-Ala-Val-Gly-Ile-Gly-Ser-Thr-Ile-Phe-Cys-Lys-Ile-Ser-Lys-Lys-Cys	24	Free radical scavenging activity: DPPH and ABTS^•+^; ABTS^•+^ and DPPH free radical scavenging kinetics.	1 Pro 2 Cys	[32]
	Brevinin-1TP3	FLPGLIKVAVGVGSTILCKITKKC Phe-Leu-Pro-Gly-Leu-Ile-Lys-Val-Ala-Val-Gly-Val-Gly-Ser-Thr-Ile-Leu-Cys-Lys-Ile-Thr-Lys-Lys-Cys	24	Free radical scavenging activity: DPPH and ABTS^•+^; ABTS^•+^ and DPPH free radical scavenging kinetics.	1 Pro 2 Cys	[32]
	Brevinin-1LF1	FLPMLAGLAANFLPKIICKITKKC Phe-Leu-Pro-Met-Leu-Ala-Gly-Leu-Ala-Ala-Asn-Phe-Leu-Pro-Lys-Ile-Ile-Cys-Lys-Ile-Thr-Lys-Lys-Cys	24	Free radical scavenging activity: DPPH and ABTS^•+^; ABTS^•+^ and DPPH free radical scavenging kinetics.	2 Pro 2 Cys	[32]
	Brevinin-1FL	FWERCSRWLLN Phe-Trp-Glu-Arg-Cys-Ser-Arg-Trp-Leu-Leu-Asn	11	Free radical scavenging activity: DPPH, ABTS^•+^, NO, FRAP; MDA, SOD, CAT, LHD, and GSH levels in PC-12 cell. ROS detection in PC-12 cell (DCFH-DA assay) flow cytometry.	2 Trp 1 Cys	[71]
Nigroain	Nigroain -B-MS1	CVVSSGWKWNYKIRCKLTGNC Cys-Val-Val-Ser-Ser-Gly-Trp-Lys-Trp-Asn-Tyr-Lys-Ile-Arg-Cys-Lys-Leu-Thr-Gly-Asn-Cys	21	Free radical scavenging activity: DPPH and ABTS^•+^; ABTS^•+^ and DPPH free radical scavenging kinetics.	2 Trp 1 Tyr 3 Cys	[72]
	Nigroain -C-MS1	FKTWKNRPILSSCSGIIKG Phe-Lys-Thr-Trp-Lys-Asn-Arg-Pro-Ile-Leu-Ser-Ser-Cys-Ser-Gly-Ile-Ile-Lys-Gly	19	Free radical scavenging activity: DPPH and ABTS^•+^; ABTS^•+^ and DPPH free radical scavenging kinetics.	1 Trp 1 Cys	[72]
	Nigroain-D-SN1	CQWQFISPSRAGCIGP Cys-Gln-Trp-Gln-Phe-Ile-Ser-Pro-Ser-Arg-Ala-Gly-Cys-Ile-Gly-Pro	16	Free radical scavenging activity: DPPH and ABTS^•+^; ABTS^•+^ and DPPH free radical scavenging kinetics.	2 Pro 1 Trp 2 Cys	[72]
	Nigroain -K-SN1	SLWETIKNAGKGFILNILDKIRCKVAGGCKT Ser-Leu-Trp-Glu-Thr-Ile-Lys-Asn-Ala-Gly-Lys-Gly-Phe-Ile-Leu-Asn-Ile-Leu-Asp-Lys-Ile-Arg-Cys-Lys-Val-Ala-Gly-Gly-Cys-Lys-Thr	31	Free radical scavenging activity: DPPH and ABTS^•+^; ABTS^•+^ and DPPH free radical scavenging kinetics.	1 Trp 2 Cys	[72]
Andersonin	Andersonin-AOP1	FLPGLECVW Phe-Leu-Pro-Gly-Leu-Glu-Cys-Val-Trp	9	Free radical scavenging activity: DPPH and ABTS^•+^; inducibility of antioxidant activities of skin secretions and skin tolerance to UVB exposure.	1 Pro 1 Trp 1 Cys	[56]
	Andersonin-AOP2	SLSCFLSFTR Ser-Lys-Ser-Cys-Phe-Leu-Ser-Phe-Thr-Arg	10	Free radical scavenging activity: DPPH and ABTS^•+^; inducibility of antioxidant activities of skin secretions and skin tolerance to UVB exposure.	1 Cys	[56]
	Andersonin-AOP10a	SYLNSLSCFLSFT Ser-Tyr-Leu-Asn-Ser-Leu-Ser-Cys-Phe-Leu-Ser-Phe-Thr	13	Free radical scavenging activity: DPPH and ABTS^•+^; inducibility of antioxidant activities of skin secretions and skin tolerance to UVB exposure.	1 Cys 1 Tyr	[56]
	Andersonin-AOP11a	GMGYMMLCGLSGM Gly-Met-Gly-Tyr-Met-Met-Leu-Cys-Gly-Leu-Ser-Gly-Met	13	Free radical scavenging activity: DPPH and ABTS^•+^; inducibility of antioxidant activities of skin secretions and skin tolerance to UVB exposure.	4 Met 1 Tyr 1 Cys	[56]
	Andersonin-AOP14d	TGKHMAGCFKFGES Thr-Gly-Lys-His-Met-Ala-Gly-Cys-Phe-Lys-Phe-Gly-Glu-Ser	14	Free radical scavenging activity: DPPH and ABTS^•+^; inducibility of antioxidant activities of skin secretions and skin tolerance to UVB exposure.	1 Cys 1 Met	[56]
	Andersonin-AOP15	AYMKQHMYCAASFF Ala-Tyr-Met-Lys-Gln-His-Met-Tyr-Cys-Ala-Ala-Ser-Phe-Phe	14	Free radical scavenging activity: DPPH and ABTS^•+^; inducibility of antioxidant activities of skin secretions and skin tolerance to UVB exposure.	2 Met 2 Tyr 1 Cys	[56]
	Andersonin-AOP16	VVKCSYRQGSPDSR Val-Val-Lys-Cys-Ser-Tyr-Arg-Gln-Gly-Ser-Pro-Asp-Ser-Arg	14	Free radical scavenging activity: DPPH and ABTS^•+^; inducibility of antioxidant activities of skin secretions and skin tolerance to UVB exposure.	1 Pro 1 Tyr 1 Cys	[56]
	Andersonin-AOP23	GLFSMILGVGKKTLCGLSGLW Gly-Leu-Phe-Ser-Met-Ile-Leu-Gly-Val-Gly-Lys-Lys-Thr-Leu-Cys-Gly-Leu-Ser-Gly-Leu-Trp	21	Free radical scavenging activity: DPPH and ABTS^•+^; inducibility of antioxidant activities of skin secretions and skin tolerance to UVB exposure.	1 Trp 1 Cys 1 Met	[56]
Odorranain	Odorranain-G-OT	FVPAILCSILKTC Phe-Val-Pro-Ala-Ile-Leu-Cys-Ser-Ile-Leu-Lys-Thr-Cys	13	Free radical scavenging activity: DPPH.	1 Pro 2 Cys	[58]
	Odorranain-A-OT	VVKCSFRPGSPAPRCK Val-Val-Lys-Cys-Ser-Phe-Arg-Pro-Gly-Ser-Pro-Ala-Pro-Arg-Cys-Lys	16	Free radical scavenging activity: DPPH.	3 Pro 2 Cys	[58]
	Odorranain-M-OM	AMRLTYNRPCIYAT Ala-Met-Arg-Leu-Thr-Tyr-Asn-Arg-Pro-Cys-Ile-Tyr-Ala-Tyr	14	Free radical scavenging activity: DPPH.	1 Pro 1 Cys 1 Met 2 Tyr	[58]
	Odorranain-A-OA11	VVKCSYRQGSPDSR Val-Val-Lys-Cys-Ser-Tyr-Arg-Gln-Gly-Ser-Pro-Asp-Ser-Arg	14	Free radical scavenging activity ABTS^•+^.	1 Pro 1 Cys 1 Tyr	[59]
Hainanenin	Hainanenin-1	FALGAVTKLLPSLLCMITRKC Phe-Ala-Leu-Gly-Ala-Val-Thr-Lys-Leu-Leu-Pro-Ser-Leu-Leu-Cys-Met-Ile-Thr-Arg-Lys-Cys	21	Free radical scavenging activity: DPPH.	1 Pro 2 Cys 1 Met	[55]
	Hainanenin-5	FALGAVTKRLPSLFCLITRKC Phe-Ala-Leu-Gly-Ala-Val-Thr-Lys-Arg-Leu-Pro-Ser-Leu-Phe-Cys-Leu-Ile-Thr-Arg-Lys-Cys	21	Free radical scavenging activity: DPPH.	1 Pro 2 Cys	[55]
FW	FW-1	FWPLI(NH_2_) Phe-Trp-Pro-Leu-Ile(NH_2_)	5	The fluorometric assay (DCFH-DA assay) to measure the intracellular ROS level.	1 Pro 1 Trp	[43]
	FW-2	FWPMI(NH_2_) Phe-Trp-Pro-Met-Ile(NH_2_)	5	The fluorometric assay (DCFH-DA assay) to measure the intracellular ROS level.	1 Pro 1 Trp 1 Met	[43]
Taipehensin	Taipehensin-1TP1	TLIWEFYHQILDEYNKENKG Thr-Leu-Ile-Trp-Glu-Phe-Tyr-His-Gln-Ile-Leu-Asp-Glu-Tyr-Asn-Lys-Glu-Asn-Lys-Gly	20	Free radical scavenging activity: DPPH and ABTS^•+^; ABTS^•+^ and DPPH free radical scavenging kinetics.	2 Tyr 1 Trp	[32]
	Taipehensin-2TP	CLMARPNYRCKIFKQC Cys-Leu-Met-Ala-Arg-Pro-Asn-Tyr-Arg-Cys-Lys-Ile-Phe-Lys-Gln-Cys	16	Free radical scavenging activity: DPPH and ABTS^•+^; ABTS^•+^ and DPPH free radical scavenging kinetics.	1 Pro 3 Cys 1 Met	[32]
OM	OM-GF17	GFFKWHPRCGEEHSMWT Gly-Phe-Phe-Lys-Trp-His-Pro-Arg-Cys-Gly-Glu-Glu-His-Ser-Met-Trp-Thr	17	Free radical scavenging activity: DPPH, ABTS^•+^, and NO; FRAP.	1 Pro 1 Met 2 Trp 1 Cys	[73]
	OM-LV20	LVGKLLKGAVGDVCGLLPIC Leu-Val-Gly-Lys-Leu-Leu-Lys-Gly-Ala-Val-Gly-Asp-Val-Cys-Gly-Leu-Leu-Pro-Ile-Cys	20	Free radical scavenging activity: DPPH, ABTS^•+^, and NO; FRAP.	1 Pro 2 Cys	[74]
	OM-GL15	GLLSGHYGRASPVAC Gly-Leu-Leu-Ser-Gly-His-Tyr-Gly-Arg-Ala-Ser-Pro-Val-Ala-Cys	15	Free radical scavenging activity: DPPH, ABTS^•+^, and FRAP.	1 Pro 1 Cys 1 Tyr	[31]
OA	OA-GL21	GLLSGHYGRVVSTQSGHYGRG Gly-Leu-Leu-Ser-Gly-His-Tyr-Gly-Arg-Val-Val-Ser-Thr-Gln-Ser-Gly-His-Tyr-Gly-Arg-Gly	21	Free radical scavenging activity: DPPH and ABTS^•+^.	2 Tyr	[75]
	OA-VI12	VIPFLACRPLGL Val-Ile-Pro-Phe-Leu-Ala-Cys-Arg-Pro-Leu-Gly-Leu	12	ROS levels in HaCaT cells treated with H_2_O_2_ or UVB irradiation (DCFH-DA assay); measurement of SOD and GSH levels in UVB-irradiated mouse skin.	2 Pro 1 Cys	[11]
Temporin	Temporin-TP1	FLPVLGKVIKLVGGLL(NH_2_) Phe-Leu-Pro-Val-Leu-Gly-Lys-Val-Ile-Lys-Leu-Val-Gly-Gly-Leu-Leu(NH_2_)	16	Free radical scavenging activity: DPPH and ABTS^•+^; ABTS^•+^ and DPPH free radical scavenging kinetics.	1 Pro	[32]
	Temporin-MS1	FLTGLIGGLMKALGK Phe-Leu-Thr-Gly-Leu-Ile-Gly-Gly-Leu-Met-Lys-Ala-Leu-Gly-Lys	15	Free radical scavenging activity: DPPH and ABTS^•+^; ABTS^•+^ and DPPH free radical scavenging kinetics.	1 Met	[72]
Triptofilins	Pat-2	FPPWL(NH_2_) Phe-Pro-Pro-Trp-Leu(NH_2_)	5	ROS and RNS intracellular analysis by flow cytometry assay. In silico antioxidant studies.	2 Pro 1 Trp	[48]
	PpT-2	FPWLLS(NH_2_) Phe-Pro-Trp-Leu-Leu-Ser(NH_2_)	6	Free radical scavenging activity: ABTS^•+^ and DPPH; ROS and RNS intracellular analysis by flow cytometry assay.	1 Pro 1 Trp 1 Ser	[76]
Wuchuanin	Wuchuanin-AOP5	TVWGFRPSKPPSGYR Thr-Val-Trp-Gly-Phe-Arg-Pro-Ser-Lys-Pro-Pro-Ser-Gly-Tyr-Arg	15	Free radical scavenging activity: ABTS^•+^.	3 Pro 1 Tyr 1 Trp	[59]
	Wuchuanin-A1	APDRPRKFCGILG Ala-Pro-Asp-Arg-Pro-Arg-Lys-Phe-Cys-Gly-Ile-Leu-Gly	13	Free radical scavenging activity: ABTS^•+^.	2 Pro 1 Cys	[59]
Ranacyclin	Ranacyclin-HB1	GAPKGCWTKSYPPQPCFGKK Gly-Ala-Pro-Lys-Gly-Cys-Trp-Thr-Lys-Ser-Tyr-Pro-Pro-Gln-Pro-Cys-Phe-Gly-Lys-Lys	20	Free radical scavenging activity: ABTS^•+^ and DPPH.	4 Pro 2 Cys 1 Trp 1 Tyr	[72]
Daiyunin	Daiyunin-1	CGYKYGCMVKVDR Cys-Gly-Tyr-Lys-Tyr-Gly-Cys-Met-Val-Lys-Val-Asp-Arg	13	Free radical scavenging activity: DPPH and ABTS^•+^; ABTS^•+^ and DPPH free radical scavenging kinetics.	1 Met 2 Tyr 2 Cys	[72]
Pleskein	Pleskein-2	FFLLPIPNDVKCKVLGICKS Phe-Phe-Leu-Leu-Pro-Ile-Pro-Asn-Asp-Val-Lys-Cys-Lys-Val-Leu-Gly-Ile-Cys-Lys-Ser	20	Free radical scavenging activity: DPPH and ABTS^•+^; ABTS^•+^ and DPPH free radical scavenging kinetics.	2 Pro 2 Cys	[72]
OS	OS-LL11	LLPPWLCPRNK Leu-Leu-Pro-Pro-Trp-Leu-Cys-Pro-Arg-Asn-Lys	11	Photodamage and mechanisms related to the ability to eliminate free radicals: ABTS^•+^, DPPH; ROS, Lipid peroxide (LPO), MDA, LDH, and CAT levels in mouse keratinocytes.	3 Pro 1 Trp	[77]
Odorranaopin	Odorranaopin-MS2	DNVYSRPPQRFGQNVIS Asp-Asn-Val-Tyr-Ser-Arg-Pro-Pro-Gln-Arg-Phe-Gly-Gln-Asn-Val-Ile-Ser	17	Free radical scavenging activity: DPPH and ABTS^•+^; ABTS^•+^ and DPPH free radical scavenging kinetics.	2 Pro 1 Tyr	[72]
Jindongenin	Jindongenin-1a	DSMGAVKLAKLLIDKMKCEVTKAC Asp-Ser-Met-Gly-Ala-Val-Lys-Leu-Ala-Lys-Leu-Leu-Ile-Asp-Lys-Met-Lys-Cys-Glu-Val-Thr-Lys-Ala-Cys	24	Free radical scavenging activity: DPPH.	2 Met 2 Cys	[34]
Parkerin	Parkerin	GWANTLKNVAGGLCKITGAA Gly-Trp-Ala-Asn-Thr-Leu-Lys-Asn-Val-Ala-Gly-Gly-Leu-Cys-Lys-Ile-Thr-Gly-Ala-Ala	20	Free radical scavenging activity: DPPH.	1 Trp 1 Cys	[33]
Hejiangin	Hejiangin-A1	RFIYMKGFGKPRFGKR Arg-Phe-Ile-Tyr-Met-Lys-Gly-Phe-Gly-Lys-Pro-Arg-Phe-Gly-Lys-Arg	16	Free radical scavenging activity: ABTS^•+^	1 Pro 1 Tyr 1 Met	[59]
APBMH	APBMH	LEQQVDDLEGSLEQEKK Leu-Glu-Gln-Gln-Val-Asp-Asp-Leu-Glu-Gly-Ser-Leu-Glu-Gln-Glu-Lys-Lys	17	Hydroxyl radical scavenging activity; superoxide radical scavenging activity; lipid peroxidation inhibition assay; protective effect of the purified peptide against hydroxyl radical-induced DNA damage.	-	[78]
Salamandrin	Salamandrin-I	FAVWGCADYRGY(NH_2_) Phe-Ala-Val-Trp-Gly-Cys-Ala-Asp-Tyr-Arg-Gly-Tyr(NH_2_)	12	Free radical scavenging activity: ABTS^•+^ and DPPH; in silico antioxidant studies.	1 Cys 1 Tyr	[79]
APBSP	APBSP	LEELEEELEGCE Leu-Glu-Glu-Leu-Glu-Glu-Glu-Leu-Glu-Gly-Cys-Glu	12	Lipid peroxidation inhibition assay; scavenging effect on DPPH radical; hydroxyl radicals scavenging activity; superoxide anion radical scavenging activity; peroxyl radicals scavenging activity.	1 Cys	[47]
W3	W3	LGWVSKGKLL(NH_2_) Leu-Gly-Trp-Val-Ser-Lys-Gly-Lys-Leu-Leu(NH_2_)	10	Free radical scavenging activity: DPPH; Fe^2+^ chelating capability.	1 Trp	[80]
Macrotympanain-A1	Macrotympanain-A1	FLPGLECVW Phe-Leu-Pro-Gly-Leu-Glu-Cys-Val-Trp	9	Free radical scavenging activity: ABTS^•+^.	1 Pro 1 Cys 1 Trp	[59]
Ansin2	Ansin2	TRCFRVCS Thr-Arg-Cys-Phe-Arg-Val-Cys-Ser	8	Free radical scavenging activity: DPPH and ABTS^•+^; FRAP.	1 Cys	[80]
Frog Protein	Hydrolysates (FPHs)	L/IK Leu/Ile-Lys	2	Free radical scavenging activity: DPPH and ORAC; FRAP.	-	[81]
	Hydrolysates (FPHs)	FK Phe-Lys	2	Free radical scavenging activity: DPPH and ORAC; FRAP.	-	[81]

## Data Availability

No new data were created or analyzed in this study.

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
