# Peer review of "The Role of Amphibian AMPs Against Oxidative Stress and Related Diseases"

_antibiotics, 2025, doi:10.3390/antibiotics14020126_

Round 1
Reviewer 1 Report
Comments and Suggestions for Authors
1. Please use the full name of AMPs in the title and specify the main finding of your review result in the Abstract not just informed generally.
2. Please make sure that "amphibians" use in this review article manuscript cover the related species of them correctly.
3. Line 109, please check the initial sentence.
4. Where is Table 1 as you mentioned in the content.
5. The photographs of all your claimed amphibian species should be included that might be initially or before Conclusion. Or as summarized table with supporting references.
6. Please set as summarized table based on bioactivities: Antioxidant, Antimicrobial, and so on as your mentioned and presented in the context.
7. Have the authors included the review on Patent and including in the content?
8. Please separate "Concluding" from "Prospects....." which the later come before Conclusion.
9. Your mention on "molecular dynamics" in Conclusion. The content presents about molecular docking. Please specify or more explain for them and confirm or add the MD result in the content.
10. Your mention based on skin AMPs, please inform and suggest about their amphibian saliva AMPs?
11. There are many reviews about AMP, please more specify to differentiate from the previous published review articles and create the presentation with more interest to the reader.
Reviewer 2 Report
Comments and Suggestions for Authors
The authors present a thorough review of the literature on the antioxidant and antimicrobial action of small peptides isolated from amphibian skin. The manuscript describes the different classes of peptides in great detail, and the authors make a good effort to summarize the data on the peptides’ protective effects against neurodegenerative diseases and cancer. The references are current, and the entire breadth of the field is covered in the review. The sheer volume of information presented makes it impossible to vouch for the accuracy of every reference, but a spot check shows that the original literature is cited accurately. A table summarizing the antioxidant and antimicrobial actions of the listed peptides would have helped, and a phylogenetic analysis of the peptides would have added to the value of the paper. But minor suggestions aside, the manuscript is informative, accurate and well-written, and would certainly be of interest to the reader of “Antibiotics”.
Minor critiques
Figure 1: The composition of the figure is confusing. Panel A shows a peptide surrounded by amino acids which apparently produces free radicals. It further shows that free radical production is inhibited by “free radical neutralization”. If the intent is to show how AOP neutralize free radicals, there are better ways to present this. Panel C is just a table of conditions. Please consider redrawing the figure.
Line 931: The sentence “In vitro…” repeats.
Reviewer 3 Report
Comments and Suggestions for Authors
Dear Author.
I read your paper with great interest, which provides a comprehensive review of the antioxidant properties of AOP and a clear explanation of its scavenging effect on ROS, as well as a good description of AOP's potential as a future drug. It was a good description of the potential of AOP as a drug for the future. The experiments were conducted both in vitro and in vivo, which I liked very much. However, I was concerned about the complexity of the description, the lack of experimental data, and the weak consideration of the medical applications of peptides. In view of these, I think further consideration is needed. Would you please consider the following points?
#1: Could you cite quantitative properties, especially in statistical studies, to reinforce the experimental results?
#2: In the discussion part, could you expand and explore the merits and demerits, or should I say the pros and cons, of the application in medicine?
#3: Could you provide a more detailed, specific, and easy-to-understand description of the experimental method with references?
#4: I found it difficult to read because of duplicated descriptions, e.g. explanations. Could this be reorganised to make it a little easier to read?
It seems to me that the above revisions would make the paper even better.
Comments on the Quality of English LanguageI think the English is average. It did not harm my understanding of the content at all. In my query to the author, I asked for cleaner language to avoid redundancy, but I think it takes a high level of linguistic ability to achieve such a good flow of English. If the author is not a native English speaker, I think it is necessary to ask for help from someone who is, but if you are an everyday user of English, I suggest that you use more refined English, make the paragraphs more connected, and be more conscious of the word ‘thesis’ style. I think it is necessary to make an effort to replace everyday conversational expressions such as ‘amazing’ and ‘very’ with scientific expressions. I think it is difficult to find the right balance between the two, but I think the impression could be changed considerably by modifying the English.
Round 2
Reviewer 1 Report
Comments and Suggestions for Authors
The ahthors have improved the quality of manuscript well a d well addressed all comments r